# Learning from the Best, Differently: A Diversity-Driven Rethinking on Data Selection

## Abstract

High-quality pre-training data is a decisive factor for large language models, where quality captures factual reliability and semantic value, and diversity ensures broad coverage and distributional heterogeneity. Existing approaches typically rely on single or multiple-dimensional score-based selection. However, empirical studies have shown that directly selecting top-scored data often degrades downstream performance, and sampling from a broader range is required to recover results. The above non-monotonicity between the dataset scores and the downstream benchmark results reveals a fundamental bias: score-based methods collapse correlated dimensions, causing top-scored data to appear high-quality while systematically overlooking diversity. We argue that ensuring diversity requires decomposing correlated evaluation metrics into orthogonal feature dimensions, from which the top-scored data can be directly selected. To this end, we proposed the **O**rthogonal **Di**versity-Aware **S**election (**ODiS**) algorithm, a method to preserve both quality and diversity during high-quality data selection. First, ODiS evaluates data from multiple dimensions, covering language quality, knowledge quality, and comprehension difficulty. The resulting multi-dimensional scores are then decorrelated via Principal Component Analysis (PCA), yielding orthogonal evaluation dimensions. For each dimension, a Roberta-based scorer is trained to regress the data onto PCA-projected scores, enabling scalable inference on large corpora. Finally, ODiS constructs the training dataset by selecting top-scored data within each orthogonal dimension, thereby ensuring both quality and diversity. Empirical results show that ODiS-selected data exhibit less than 2% inter-dimension overlap, confirming the orthogonality between dimensions. More importantly, models trained with ODiS-selected data significantly outperform other baselines on multiple downstream benchmarks, highlighting the necessity of orthogonal, diversity-aware data selection for LLMs.

## 1 Introduction

Pretraining is the primary stage for models to acquire fundamental abilities, such as language understanding, text generation and information extraction (Brown et al., 2020; Chowdhery et al., 2023; Roberts et al., 2020). These capabilities are largely determined by the quality and diversity of the training data. Quality captures authenticity, reliability, and semantic integrity, ensuring that models learn accurate and well-structured knowledge. Diversity, on the other hand, emphasizes coverage and comprehensiveness, enabling models to generalize across domains and tasks. With the increase of both model and corpus sizes, designing efficient data selection methods that jointly account for these two aspects has become a critical challenge for advancing model performance.

Existing works have proposed diverse methods for selecting data based on quality and diversity. Quality-based methods typically utilize rule-based heuristics (Laurençon et al., 2022; Weber et al., 2024; Penedo et al., 2023; Raffel et al., 2020; Lee et al., 2021), such as document length constraint and content deduplication, or score-based techniques (Wenzek et al., 2019; Touvron et al., 2023; Wettig et al., 2024; Penedo et al., 2024; Su et al., 2024), where classifiers or perplexity models assign a single quality score to filter noisy or irrelevant data. Diversity-based methods (He et al., 2024; Zhang et al., 2024; Tirumala et al., 2023; Yang et al., 2025) instead focus on broadening coverage by mixing data across domains or clustering in the embedding space, thereby reducing redundancy and expanding the distribution. Recent works also attempt to combine quality and diversity to select

data (Zhuang et al., 2025; Liu et al., 2025; Bai et al., 2025), typically formulating both diversity and quality into a multi-dimensional score or mixing selected data from different domains. However, the intrinsic correlation between dimensions makes weight tuning challenging, and naive aggregation often results in overlapping signals. In summary, current works encounter three key challenges: **(1)** The data selected with top scores are not always optimal, and sampling is necessary to achieve satisfactory performance, but the cause remains unexplored. **(2)** Scored-based methods combine multiple aspects into a one-dimensional signal, making them unable to capture both quality and diversity. **(3)** Delicate hyper-parameter tuning is required to balance the influence from different dimensions, undermining generality and practical deployment of the methods.

To address these challenges, we first revisit the problem of data selection through the lens of bias and correlation, identifying the neglect of diversity as the fundamental cause. Guided by the insight, we propose **O**rthogonal **Di**versity-Aware **S**election (ODiS) algorithm, a method to effectively construct a dataset with both quality and diversity. Specifically, drawing inspiration from (Zhuang et al., 2025; Wettig et al., 2024), we label a reference dataset from 11 dimensions, covering four main categories: language quality, knowledge quality, comprehension difficulty, and information quality. After analyzing the correlation across the dimensions, we reveal strong entanglement between them, which will introduce redundancy and bias the score-based selection. To mitigate this, we apply Principal Component Analysis (PCA) to scores and derive orthogonal evaluation dimensions. For acceleration and scalability, Roberta-based scorers are trained to predict scores along each PC dimension on the target dataset. Finally, ODiS constructs the training set by selecting the top samples from orthogonal PC dimensions, ensuring both quality and diversity of the dataset.

Empirical validations demonstrate that the model trained on data selected by the proposed ODiS methods achieves the best in various downstream benchmarks, compared with existing baselines DSIR, PPL, and Nemotron-CC. We analyze the source of performance gains by comparing top-scored data with samples drawn from broader ranges of each dimension, and find that top-scored subsets consistently underperform, whereas combining data across dimensions substantially improves performance. Data analysis further confirms the strong dimension correlations before PCA and demonstrates that orthogonal principal components capture distinct aspects of the data, thereby validating the data quality and diversity in the selected dataset. Finally, the ablation study suggests that increasing the number of dimensions will marginally improve performance, indicating an efficiency-performance trade-off.

The main contributions of this work are as follows: **(1)** We provide the first analysis about performance degradation of top-scored data, identifying neglected diversity as the underlying cause. **(2)** We propose the ODiS algorithm, a score-based data selection algorithm that explicitly ensures both quality and diversity through dimension decomposition. **(3)** The analysis results reveal that ODiS benefits from dimension decomposition and enhanced data diversity, which sheds light on future data selection methods for reducing inter-dimensional correlations to improve data diversity.

## 2 METHOD

### 2.1 SCORE-BASED BIAS IN DATA SELECTION

We begin by analyzing the causes of selection bias and the role of data sampling through examining model performance with varying data sizes. Following Wettig et al. (2024), we establish multiple metrics targeting different semantic features, whose details are provided in Section 2.3. Then, we utilize a score-based method to filter candidate pools of varying sizes from the Nemotron-CC dataset (Su et al., 2024), selecting data based on the average scores from the above metrics. Specifically, we select the data with the top-k scores at different scales, ranging from 100B to 900B tokens. After that, we train a 1.5B-parameter model from scratch on these subsets, with 100B tokens training data budget. Finally, model performance is evaluated across multiple benchmarks. The training details and benchmark selection are provided in 3.1.

From Figure 1, we can observe that the data with the highest score (e.g., the candidate pool size equals 100B) performs the worst, while sampling data from a broader range (e.g., the candidate pool size larger than 100B) leads to improvements. This reveals a non-monotonic relationship between the data score and model performance, which complicates data selection: the sampling range and other potential hyperparameters should be optimized accordingly. Since the top-scored data has

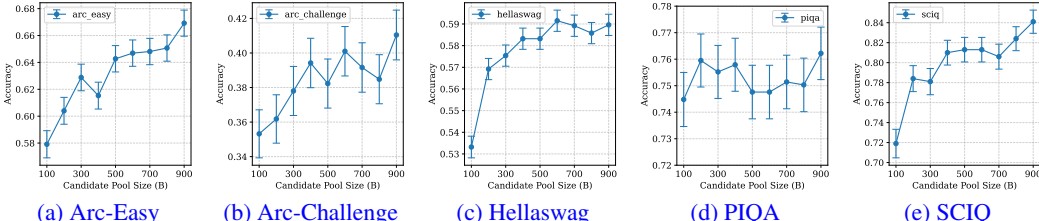

(a) Arc-Easy    (b) Arc-Challenge    (c) Hellaswag    (d) PIQA    (e) SCIQ

Figure 1: Performance comparison of 1.5B models trained on a fixed 100B token budget randomly sampled from candidate pools of varying sizes. The x-axis denotes the Top-K candidate pool size ranked by average scores. The leftmost point (100B) represents training on the highest-scored data, while moving right implies including lower-scored data.

the highest quality, we further investigate its diversity through an embedding-based visualization to determine the reason behind the non-monotonicity.

We sample data from the above candidate pools and visualize the UMAP projection of text embeddings together with the distribution of pairwise distances. As shown in Figure 2a, the purple points (Top-100B) tend to cluster in a limited region, while the yellow points (Top-500B) span a broader space. This demonstrates that directly selecting the data with the highest score results in more uncovered space, whereas broader selection achieves comprehensive semantic coverage. Moreover, Figure 2b demonstrates that as the selection scope increases (e.g., from 100B to 500B), the pairwise distance distribution shifts rightward, indicating an increase in semantic diversity. These results suggest that the increase in data size widens the distinction between data and amplifies data diversity, whereas top-scored data is relatively homogeneous, which explains the performance degradation of the top-scored data.

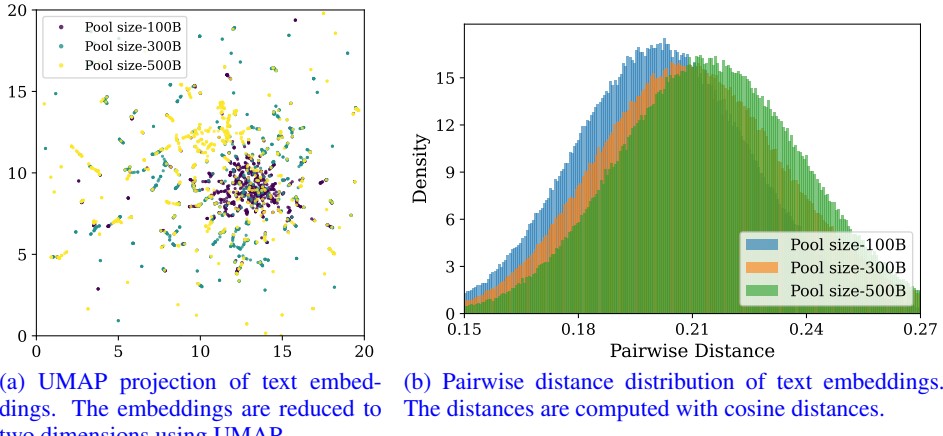

(a) UMAP projection of text embeddings. The embeddings are reduced to two dimensions using UMAP.

(b) Pairwise distance distribution of text embeddings. The distances are computed with cosine distances.

Figure 2: Visualization of data embeddings from different candidate pool sizes. The embeddings are generated using 1000 randomly sampled texts with the m3e-base model (Wang Yuxin, 2023), followed by L2 normalization and removal of the top 3 principal components to suppress dominant common directions. The pool size indicates the number of top-ranked tokens from the dataset.

## 2.2 PROBLEM FORMULATION

Motivated by the previous analysis in Section 2.1, we focus on enhancing data diversity while ensuring data quality during data selection. Our objective is to select the most valuable subset of a large corpus to facilitate the model pre-training. Instead of directly processing the target dataset $\mathcal{D}_t$, we first take a smaller reference dataset $\mathcal{D}_r$ to generate the scoring strategy. Specifically, each data $x_i$ in the reference dataset $\mathcal{D}_r = \{x_i\}_{i=1}^N$ will be labeled from $m$ dimensions, whose score vector can be expressed as $\boldsymbol{\alpha}^{(i)} = (\alpha_1^{(i)}, \alpha_2^{(i)}, \cdots, \alpha_m^{(i)})$. The goal is to design a mapping function $F(\cdot)$ that transforms the scores $\boldsymbol{\alpha}^{(i)}$ into a ranking, through which we can directly select the top-scored data. The function should be designed such that training on the top-scored data maximizes the performance of

downstream tasks. The problem can be written as follows:

$$\mathcal{D}_s = \arg \max_{\mathcal{D}_s \subset \mathcal{D}_t} G(\mathcal{D}_s), \tag{1}$$

where $G(\cdot)$ denotes evaluation performance on benchmarks tasks.

## 2.3 MULTI-DIMENSION DATA EVALUATION

Instead of directly optimizing the data selection for downstream tasks, which may be biased toward task-specific signals, we propose a method that focuses on enhancing data quality and diversity for general purposes. To comprehensively evaluate each data document, we set up 11 dimensions, i.e., $m = 11$, for four general aspects: Language quality, Knowledge quality, Comprehension difficulty, and Information Quality. Without loss of generality, we still use $m$ to denote the number of dimensions in the algorithm. We briefly describe the dimensions as follows, and the details are in Appendix B:

- **Language quality.** We prefer data that is (i) coherent in structure, (ii) concise without redundancy, and (iii) correct in spelling/grammar and word choice (Penedo et al. (2023)).

- **Knowledge quality.** We value content with (i) sufficient coverage and (ii) depth, (iii) useful reasoning signals, (iv) clear educational, and (v) practical value (Gunasekar et al. (2023); Guo et al. (2025)).

- **Comprehension difficulty.** We assess the difficulty level, i.e., conceptual complexity and domain professionalism, as higher difficulty can improve generalization (Agrawal & Singh (2023)).

- **Information quality.** We require (i) factual accuracy and (ii) sufficient completeness so models can learn reliable, fully specified facts (Chang et al. (2024)).

Each document is assigned a score from 0 to 5 on every dimension using the OpenAI GPT API. Detailed definitions of the metrics and prompt are provided in Appendix C. The resulting scores constitute a matrix $\mathbf{X}$:

$$\mathbf{X} = \begin{bmatrix} \alpha_1^{(1)} & \cdots & \alpha_m^{(1)} \\ \vdots & \ddots & \vdots \\ \alpha_1^{(N)} & \cdots & \alpha_m^{(N)} \end{bmatrix} \in \mathbb{R}^{N \times m}. \tag{2}$$

## 2.4 ORTHOGONAL DIVERSITY-AWARE SELECTION

**Dimension decomposition via PCA.** Previous studies have shown that different dimensions often exhibit correlations(Zhuang et al. (2025)), i.e., the data with higher knowledge depth may have less knowledge richness, while the data with high educational value usually achieves high information quality. Such redundancy in the raw labeled score reduces effective data diversity and hinders data selection. Therefore, instead of directly using the raw labeled scores, we will transform the scores to eliminate the potential correlation between different dimensions, which is done through principal component analysis (PCA).

To eliminate the scale difference, we first calculate the mean of each dimension $\boldsymbol{\mu}$, and obtain the data matrix $\mathbf{X}_c$ centered with the mean:

$$\boldsymbol{\mu} = \frac{1}{N} \sum_{i=1}^{N} \boldsymbol{\alpha}^{(i)}, \mathbf{X}_c = \mathbf{X} - \boldsymbol{\mu}. \tag{3}$$

After that, we compute the covariance matrix $\boldsymbol{\Sigma}$ and adopt eigen decomposition to $\boldsymbol{\Sigma}$:

$$\boldsymbol{\Sigma} = \boldsymbol{V}\boldsymbol{\Lambda}\boldsymbol{V}^T, \boldsymbol{\Sigma} = \frac{1}{N-1}\mathbf{X_c}^T\mathbf{X_c} \in \mathbb{R}^{m \times m}, \tag{4}$$

where $\boldsymbol{V} = [\mathbf{v}_1, \cdots, \mathbf{v}_m]$ are orthogonal eigenvectors and $\boldsymbol{\Lambda} = \mathrm{diag}(\lambda_1 \geq \cdots \lambda_m \geq 0)$ are eigenvalues. Each eigenvector represents an orthogonal combination of the original metrics, with the eigenvalue quantifying the variance contribution, i.e., the proportion of the feature representation that the eigenvector accounts for. The higher eigenvalue indicates that the data spreads more along this direction, and the eigenvector contains a more representative feature.

---

**Algorithm 1** Orthogonal Diversity-Aware Selection

---

**Input:** Reference dataset $\mathcal{D}_r$, target dataset $\mathcal{D}_t$, dimensions for evaluation, data budget $s$

**Output:** Selected dataset $\mathcal{D}_s$

1: For each $x_i \in \mathcal{D}_r$, obtain the $m$ dimensional score vector $\boldsymbol{\alpha}^{(i)} = (\alpha_1^{(i)}, \alpha_2^{(i)}, \cdots, \alpha_m^{(i)})$;

2: Compute the mean vector $\boldsymbol{\mu}$ and construct the centered matrix $\boldsymbol{X}_c$ as equation 3;

3: Compute the covariance matrix $\boldsymbol{\Sigma}$, and perform eigendecomposition to obtain eigenvalues $\lambda_i$ and eigenvectors $\mathbf{v}_i$;

4: Determine the number of principal components (PCs) $K$ with the threshold $\tau$;

5: Construct the project matrix $\boldsymbol{W}_K$, and project scores into the orthogonal space $\boldsymbol{\beta}^{(i)} = \mathbf{W}_K^T(\boldsymbol{\alpha}^{(i)})^T \in \mathbb{R}^K$;

6: Allocate budget $s_k$ to each PC dimension such that $\sum_{k=1}^{K} s_k = s$;

7: **for** $k = 1, \ldots, K$ **do**

8:     Train a RoBERTa-based scorer $r_k(\cdot) \in [0, 5]$ by regressing text $x_i \in \mathcal{D}_r$ to the PCA-transformed scores $\beta_k^{(i)}$;

9:     Apply $r_k(\cdot)$ to obtain predicted scores $\{\theta_k^{(i)}\}$ for all $x_i \in \mathcal{D}_t$;

10:     Given the budget $s_k$, determine threshold $t_k$ and select $\mathcal{D}_s^k = \{x_i \mid \theta_k^{(i)} > t_k, x_i \in \mathcal{D}_t\}$;

11: **end for**

12: Construct the final selected dataset $\mathcal{D}_s = \cup_{k=1}^{K} \mathcal{D}_s^k$.

---

**Score transformation.** To reduce cost and improve efficiency, we take the first $K$ principal components (PC) to satisfy the explained-variance ratio, rather than all the PCs: $\frac{\sum_{k=1}^{K} \lambda_k}{\sum_{k=1}^{m} \lambda_k} \geq \tau$, where $\tau$ is the threshold. Then, we can obtain the project matrix: $\mathbf{W}_K = [\mathbf{v}_1, \cdots, \mathbf{v}_K] \in \mathbb{R}^{m \times K}$. Through projection, the compressed score vector for $x_i$ is calculated as: $\boldsymbol{\beta}^{(i)} = \mathbf{W}_K^T(\boldsymbol{\alpha}^{(i)})^T \in \mathbb{R}^K$. Note that the principal component is a linear combination of the original dimensions, and it is difficult to interpret its meaning. Since the dimensions are orthogonal, we have decomposed each metric, and the score in each principal component represents the quality in this new dimension.

**Roberta-enabled model-based scorer.** To enhance labeling accuracy and capture the semantic feature of the principal component, we train a Roberta-based scorer $r_k(\cdot)$ to map the original text to the PCA-derived score for each PC dimension. The scorer $r_k(\cdot)$ will regress from the data $x_i$ from the reference dataset and the transformed score $\beta_k^{(i)}$. The Roberta-based scorer enables efficient inference on unseen data, preserves semantic richness, and provides a unified and noise-robust scoring framework for large-scale data selection. We then label the target dataset $\mathcal{D}_t$ with scorer $r_k(\cdot)$ to obtain the scores $\theta_k^{(i)}, k \in 1, \cdots, K, i \in \{1, \cdots, |\mathcal{D}_t|\}$. The details of the parameter setting and the training of Roberta-based scorers can be found in Appendix D.

**Dataset construction based on scores.** We allocate a data budget $s_k$ to each PC dimension considering its contribution and the total data budget: $s = \sum_{k=1}^{K} s_k$. For each dimension, a score threshold $t_k$ is set based on $s_k$, and the corresponding subset is denoted as $\mathcal{D}_s^k$. Since labeling and scorer training inevitably introduce noise, the top-scored data across each orthogonal dimension may still overlap, and directly merging the subsets will result in duplication. To address this, we apply a joint score threshold $\boldsymbol{t} = (t_1, \cdots, t_K)$ and select a data within the target dataset $x_i \in \mathcal{D}_t$ if $\theta_k^{(i)} > t_k, \forall k \in 1, \cdots, K$. The final selected data is obtained as the union: $\mathcal{D}_s = \cup_{k=1}^{K} \mathcal{D}_s^k$. This construction ensures that each dimension contributes its highest-quality data, while the orthogonality of PC dimensions guarantees enhanced diversity in the resulting dataset.

# 3 EXPERIMENTS

## 3.1 EXPERIMENT SETUP

**Dataset.** We use Nemotron-CC dataset (Su et al., 2024) as the data pool for selection, which is a large-scale dataset for pretraining large language models. The dataset comprises both real-world and synthetic data, covering major domains, such as web knowledge and question answering. For

tokenization, we use the LLaMA-3-8B tokenizer, which has a vocabulary size of 128,256, and set the maximum sequence length to 4096.

**Evaluation.** We evaluate the model performance with lm-eval-harness framework (Gao et al., 2024). We first monitor task-level performance fluctuations across training steps, with detailed results presented in Appendix F. Based on the above result, we follow the "fine task" metric (Kydlíček et al.) to select downstream tasks with performance that varies significantly as training progresses. We select five tasks covering main categories: **General Knowledge** (including Arc-Easy/Challenge (Clark et al., 2018)), **Commonsense Reasoning** (including Hellaswag (Zellers et al., 2019), SCIQ (Johannes Welbl, 2017)), and **Physical Reasoning** (PIQA (Bisk et al., 2020)).

**Training.** Each experiment is conducted under a data budget of 100B tokens. Unless specified in the caption, the *top* selection represents directly selecting a 100B token dataset for training, while *sample* selection represents selecting a 700B token dataset and sampling training data from it. We employ decoder-only models with 1.5B parameters and train them from scratch with the selected data. Training uses a global batch size of 512 and the AdamW optimizer (Loshchilov & Hutter, 2019), with a peak learning rate of 3e-4, cosine decay scheduling, and linear warmup. Unless specified, the proposed ODiS method selects data from the first 4 PC dimensions and allocates the data budget evenly to each dimension.

## 3.2 MAIN RESULTS

The main results are summarized in Table 1. We compare the proposed ODiS method with existing baselines, including DSIR (Xie et al., 2023), PPL (Ankner et al., 2024), and Nemotron-CC (Su et al., 2024), and methods adopting a similar separate-then-select paradigm, including Web-Organizer Wettig et al. (2025), Semdedup Abbas et al. (2023), and selecting directly from the original 11 dimensions. The results show that the model trained on data selected by ODiS achieves the highest performance compared with the baselines. Specifically, ODiS achieves a generally 3-point marginal improvement compared with the random sampling in average accuracy. Notably, it surpasses all the methods across all task categories, highlighting its versatility in addressing a wide range of downstream tasks. The performance gains are typically obvious in Arc-E and Arc-C, indicating their enhancement in general knowledge and content diversity. In contrast, baseline methods such as PPL and DSIR emphasize only data quality, which limits diversity and hampers overall performance. Moreover, compared with other methods that employ a similar paradigm, ODiS benefits from effective orthogonal dimension construction, which avoids the interference from distinct dimensions. The existing methods suffers from the correlation across different domains, leading to performance degradation. These results demonstrate the importance of data diversity during data selection, and decomposition is a direction for efficient diversity improvement.

| Method | Arc-C | Arc-E | Hellaswag | SCIQ | PIQA | Average |
|---|---|---|---|---|---|---|
| Random | $35.0_{\pm 1.4}$ | $62.7_{\pm 1.0}$ | $58.3_{\pm 0.5}$ | $85.5_{\pm 1.1}$ | $74.5_{\pm 1.0}$ | $63.2_{\pm 1.0}$ |
| Nemotron-HQ | $37.3_{\pm 1.4}$ | $64.6_{\pm 1.0}$ | $57.7_{\pm 0.5}$ | $83.9_{\pm 1.1}$ | $73.6_{\pm 1.0}$ | $63.4_{\pm 1.0}$ |
| PPL-*Top* | $37.9_{\pm 1.4}$ | $62.8_{\pm 1.0}$ | $54.7_{\pm 0.5}$ | $83.4_{\pm 1.2}$ | $74.7_{\pm 1.0}$ | $62.7_{\pm 1.0}$ |
| PPL-*Sample* | $36.1_{\pm 1.4}$ | $64.3_{\pm 1.0}$ | $58.4_{\pm 0.5}$ | $85.8_{\pm 1.1}$ | $74.8_{\pm 1.0}$ | $63.9_{\pm 1.0}$ |
| DSIR | $27.8_{\pm 1.3}$ | $48.5_{\pm 1.0}$ | $54.6_{\pm 0.5}$ | $78.5_{\pm 1.3}$ | $71.0_{\pm 1.1}$ | $56.1_{\pm 1.0}$ |
| PC Aver-*Top* | $35.3_{\pm 1.4}$ | $57.9_{\pm 1.0}$ | $53.3_{\pm 0.5}$ | $71.9_{\pm 1.4}$ | $74.5_{\pm 1.0}$ | $58.6_{\pm 1.1}$ |
| PC Aver-*Sample* | $39.2_{\pm 1.4}$ | $64.8_{\pm 1.0}$ | $58.9_{\pm 0.5}$ | $80.6_{\pm 1.3}$ | $75.1_{\pm 1.0}$ | $63.7_{\pm 1.0}$ |
| Web-Org *Format* | $36.4_{\pm 1.4}$ | $64.5_{\pm 1.0}$ | $57.5_{\pm 0.5}$ | $84.6_{\pm 1.1}$ | $73.7_{\pm 1.0}$ | $63.3_{\pm 1.0}$ |
| Web-Org *Topic* | $36.2_{\pm 1.4}$ | $62.1_{\pm 1.0}$ | $60.3_{\pm 0.5}$ | $83.7_{\pm 1.2}$ | $75.8_{\pm 1.0}$ | $63.6_{\pm 1.0}$ |
| Ori-11-Dim | $41.0_{\pm 1.4}$ | $66.5_{\pm 1.0}$ | $54.8_{\pm 0.5}$ | $86.5_{\pm 1.1}$ | $73.4_{\pm 1.0}$ | $64.5_{\pm 1.0}$ |
| Semdedup | $39.6_{\pm 1.4}$ | $66.4_{\pm 1.0}$ | $57.8_{\pm 0.5}$ | $85.0_{\pm 1.1}$ | $74.4_{\pm 1.0}$ | $64.6_{\pm 1.0}$ |
| **ODiS** | $\mathbf{41.6}_{\pm 1.4}$ | $\mathbf{66.9}_{\pm 1.0}$ | $\mathbf{58.4}_{\pm 0.5}$ | $\mathbf{85.6}_{\pm 1.0}$ | $\mathbf{77.4}_{\pm 1.0}$ | $\mathbf{66.0}_{\pm 1.0}$ |

Table 1: Performance across data selection methods. The results are reported as percentage accuracy, with the ± standard error shown in a smaller font size. The random selection method samples data from the whole Nemotron-CC dataset, while the Nemotron-HQ method samples from the Nemotron-CC HQ subset. The PC Aver baseline utilizes the averaged scores from PC1 to PC4. The Ori-11-Dim indicates the data selected uniformly from the original 11 dimensions.

### 3.3 ANALYSIS RESULT

#### 3.3.1 INSPECTION OF DATA BIAS

**ODiS mitigates data bias.** Figure 3a shows that models trained with top-scored data within a single dimension consistently underperform, while sampling from a broader score range yields significant performance gains. This observation highlights the inherent bias in relying exclusively on single-dimension scoring, where overemphasis on one metric neglects complementary aspects of data quality and diversity. Furthermore, Figure 3b illustrates that averaging scores from multiple dimensions only partially alleviates the data bias issue, as correlation across dimensions discussed in Section 3.3.2 continues to bias the data selection. In contrast, ODiS decomposes the dimensions into orthogonal components and selects high-quality data from each dimension, effectively mitigating data bias and enhancing diversity. Notably, ODiS achieves superior performance with the smallest subset of data, avoiding both excessive sampling ranges and unnecessary data waste.

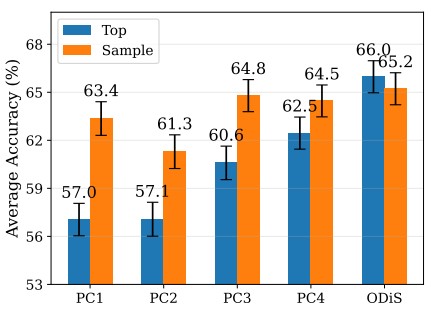
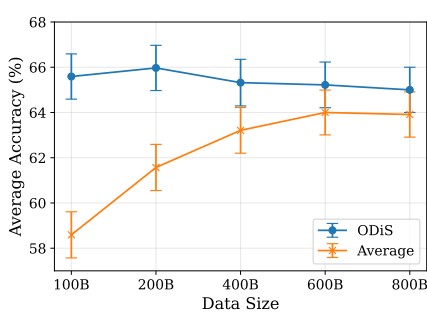

(a) Performance with data selected from PCs      (b) Performance across selected data pools

Figure 3: Model performance under different selection methods and ranges.

**ODiS effectively enhances data diversity.** To validate the diversity gain from ODiS, we compare the ODiS-selected data against score-based baselines. Figure 4 shows that the data selected by ODiS has a significantly larger average pairwise distance, even compared with a larger selected data pool, indicating enhanced data diversity. Similarly, UMAP visualization reveals that the ODiS-selected data spans a wider region in the compressed space, while the data selected with baseline methods tend to cluster around a narrower region. These results suggest that ODiS reduces redundancy and captures a wider range of semantic features during data selection.

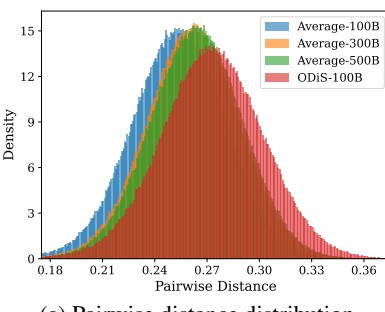
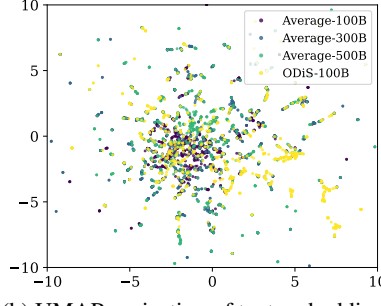

(a) Pairwise distance distribution      (b) UMAP projection of text embeddings

Figure 4: Data diversity visualization from different selection methods.

#### 3.3.2 CORRELATION BETWEEN DIMENSIONS

**Correlation of original dimensions.** Figure 5a describes the correlation coefficient across 11 dimensions with scores using 460k examples from Fineweb-Edu dataset. We observe that most dimensions exhibit weak correlation (correlation coefficient $< 0.5$), suggesting that the metrics capture distinct aspects of the data. Nonetheless, we can still discover moderate correlation between the dimensions, such as knowledge depth vs. knowledge richness, and completeness vs. knowledge richness, indicating partial overlaps in their coverage. Moreover, nearly all the dimensions exhibit at

least some degree of correlation with one another. These results suggest that although different metrics may appear conceptually orthogonal from a human perspective, they are not strictly independent in practice or from the model's viewpoint, validating the necessity of dimension decomposition.

**Correlation between original dimensions and principal components.** After applying PCA to transform dimensions into orthogonal principal components, Figure 5b describes the correlations between PC dimensions and original dimensions. Each PC exhibits a strong correlation with a subset of the original dimensions, indicating that they can represent meaningful information or semantic features. For example, PC1 aligns strongly with knowledge quality and comprehension difficulty, highlighting its central role in characterizing overall data quality. By contrast, language-related dimensions exhibit weaker direct alignment with the leading PCs. This observation indicates that the linguistic factors may already be embedded within other correlated dimensions, e.g., knowledge quality, and thus they do not emerge as dominant signals in the first few components. More generally, the orthogonal principal components are linear combinations of the original dimensions. They should not be interpreted as single semantic dimensions, but rather as abstract features that combine multiple correlated attributes and capture salient aspects of the dataset from the model's perspective.

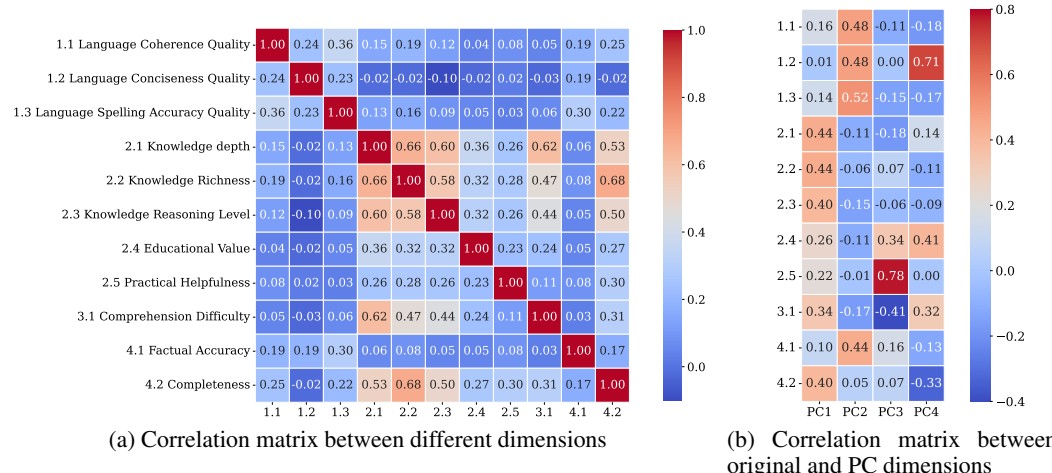

(a) Correlation matrix between different dimensions

(b) Correlation matrix between original and PC dimensions

Figure 5: Correlation analysis of evaluation dimensions. The correlations are calculated using 460k samples from the Fineweb-Edu dataset. Pairwise correlation coefficients are visualized as a heatmap.

### 3.3.3 ORTHOGONALITY BETWEEN DIMENSIONS

To assess the distinctiveness of different PC dimensions, we visualize the embeddings of top-scored data from each dimension using UMAP projection and an Upset plot, as shown in Figure 6. The UMAP visualization in Figure 6a reveals that samples from different dimensions occupy separable regions, which validates the orthogonality between dimensions and suggests that combining data from different dimensions can yield complementary data subsets. Figure 6b illustrates the marginal intersection of the data from different PC dimensions, with an overlapping ratio of 2% after tokenization, further confirming the effectiveness of the dimension decomposition. However, the data selected from the 11 original dimensions suffers from about 50% duplication.

### 3.3.4 SCALING WITH MORE DIMENSIONS

To examine whether increasing the number of PCs improves performance, we conduct an ablation study with varying PC dimensions, as summarized in Table 2. The results show that the performance improves steadily up to four dimensions, after which the performance gain becomes saturated. Beyond four PCs, additional dimensions contribute marginally to the model performance, likely because lower-variance PCs contain less information or noisy signals. Additionally, selecting more PCs will introduce increased computational cost, as labeling a large corpus is both computationally and time-consuming. These findings suggest that a small set of carefully selected orthogonal dimensions is sufficient for robust data selection, striking a balance between efficiency and effectiveness.

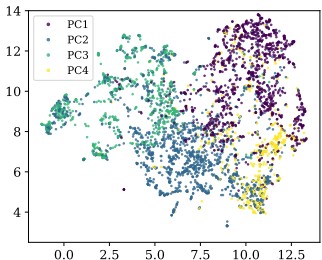

(a) UMAP for embeddings. The embeddings are generated through 1000 sampled top-scored data from each PC dimension.

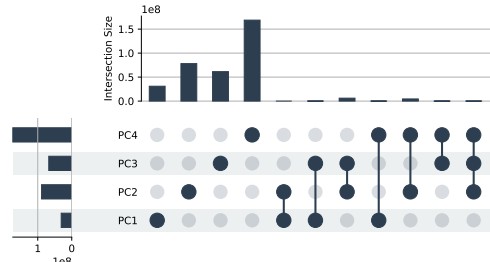

(b) UpSet plot of top-scored subsets. Data selected with top scores of each dimension are shown, with intersections indicating data that simultaneously belong to multiple top-scored subsets.

Figure 6: Data orthogonality from different PC dimensions. The data is top-scored tokens from each PC dimension, which constitutes a 100B tokens dataset.

| Method | Arc-C | Arc-E | Hellaswag | SCIQ | PIQA | Average |
|--------|-------|-------|-----------|------|------|---------|
| PC1 | $36.4_{\pm 1.4}$ | $60.4_{\pm 1.0}$ | $43.1_{\pm 0.5}$ | $81.1_{\pm 1.2}$ | $64.4_{\pm 1.1}$ | $57.1_{\pm 1.1}$ |
| PC1-2 | $35.8_{\pm 1.4}$ | $61.1_{\pm 1.0}$ | $53.7_{\pm 0.5}$ | $83.0_{\pm 1.2}$ | $75.1_{\pm 1.0}$ | $61.7_{\pm 1.0}$ |
| PC1-3 | $36.4_{\pm 1.4}$ | $60.9_{\pm 1.0}$ | $59.4_{\pm 0.5}$ | $79.1_{\pm 1.3}$ | $77.5_{\pm 1.0}$ | $62.6_{\pm 1.0}$ |
| PC1-4 | $40.5_{\pm 1.4}$ | $65.7_{\pm 1.0}$ | $57.4_{\pm 0.5}$ | $87.2_{\pm 1.1}$ | $77.2_{\pm 1.0}$ | $65.6_{\pm 1.0}$ |
| PC1-5 | $39.4_{\pm 1.4}$ | $66.9_{\pm 1.0}$ | $56.9_{\pm 0.5}$ | $88.0_{\pm 1.0}$ | $76.7_{\pm 1.0}$ | $65.6_{\pm 1.0}$ |
| **PC1-6** | $\mathbf{42.5}_{\pm 1.4}$ | $\mathbf{71.0}_{\pm 0.9}$ | $\mathbf{56.1}_{\pm 0.5}$ | $87.2_{\pm 1.0}$ | $76.0_{\pm 1.0}$ | $\mathbf{66.5}_{\pm 1.0}$ |

Table 2: Performance across different Principal Component (PC) subsets. The results are reported as percentage accuracy, with the $\pm$ standard error shown in a smaller font size.

### 3.3.5 SCALING WITH DIFFERENT MODEL SIZES AND DATA BUDGETS

To validate the scalability and generalization of our method, we conduct additional experiments with model sizes of 400M, 1.5B, and 8B, under token budgets of 100B, 200B, and 25B, respectively. The results are summarized in Table 3. We can observe that ODiS achieves consistently the best performance across these settings. The performance gain is more pronounced on PIQA, indicating an enhanced ability in the physical commonsense reasoning. Compared with the Nemotron-HQ baseline, the 400M model exhibits a larger margin on Hellaswag, while the 8B model outperforms on Arc-C and Arc-E. These results demonstrate that the proposed ODiS scales effectively across different models and data budgets, enabling more efficient training across diverse training settings.

## 4 RELATED WORKS

With the increasing scale of both model and corpora size, there is a growing demand for efficient methods to select high-quality pretraining data. The quality and diversity are two key considerations during data selection. Existing data selection methods have been proposed to enhance data quality and diversity through three main directions: non-classifier-based methods, single-classifier-based methods, and multi-classifier-based methods.

**Non-classifier-based methods.** Various works have relied on rule-based filtering with explicit heuristics or deterministic criteria (Laurençon et al., 2022; Weber et al., 2024; Penedo et al., 2023; Raffel et al., 2020; Lee et al., 2021), including language identification, URL blocks, content de-duplication, and document length thresholds. Moreover, these approaches are combined into multi-stage pipelines that sequentially perform cleaning, deduplication, quality, and safety filtering (Nguyen et al., 2023). While rule-based methods effectively improve data quality and reduce noisy data, they fail to inspect semantic-level information and will introduce distribution bias.

**Single-classifier-based methods.** In contrast to the rule-based method, it utilized a learned scoring function or discriminator to label and filter out high-quality data (Wenzek et al., 2019; Touvron et al., 2023; Wettig et al., 2024; Penedo et al., 2024; Su et al., 2024; Wang et al., 2025). Among them, language modeling perplexity has been adopted to identify data with high quality (Wenzek

| Model | Method | Arc-C | Arc-E | Hellaswag | SCIQ | PIQA | Average |
|---|---|---|---|---|---|---|---|
| **400M** | Nemotron-HQ | 29.6± 1.3 | 54.3± 1.0 | 40.7± 0.5 | 75.4± 1.4 | 67.9± 1.1 | 53.6± 1.1 |
| | PPL-*sample* | 28.2± 1.3 | 50.7± 1.0 | 41.7± 0.5 | 76.3± 1.4 | 69.9± 1.1 | 53.4± 1.1 |
| | DSIR | 24.6± 1.3 | 40.7± 1.0 | 36.5± 0.5 | 66.6± 1.5 | 65.0± 1.1 | 46.7± 1.1 |
| | PC-Aver-*Top* | 29.2± 1.3 | 48.7± 1.0 | 41.6± 0.5 | 69.9± 1.5 | 70.5± 1.1 | 52.0± 1.1 |
| | PC-Aver-*Sample* | 30.1± 1.3 | 53.2± 1.0 | 43.2± 0.5 | 75.0± 1.4 | 70.1± 1.1 | 54.3± 1.1 |
| | **ODiS** | **30.0**± 1.3 | **54.3**± 1.0 | **42.5**± 0.5 | **74.5**± 1.4 | **72.5**± 1.0 | **54.8**± 1.1 |
| **1.5B** | Nemotron-HQ | 40.3± 1.4 | 67.9± 1.0 | 59.9± 0.5 | 74.5± 1.0 | 87.1± 1.1 | 65.9± 1.0 |
| | PPL-Top | 37.4± 1.4 | 63.9± 1.0 | 56.4± 0.5 | 74.2± 1.0 | 84.5± 1.2 | 63.3± 1.0 |
| | PPL-*Sample* | 38.2± 1.4 | 67.5± 1.0 | 60.5± 0.5 | 75.6± 1.0 | 88.8± 1.0 | 66.1± 1.0 |
| | DSIR | 30.2± 1.3 | 50.8± 1.0 | 58.1± 0.5 | 72.2± 1.1 | 79.4± 1.3 | 58.1± 1.0 |
| | PC Aver-*Top* | 36.6± 1.4 | 59.1± 1.0 | 54.8± 0.5 | 74.9± 1.0 | 72.8± 1.4 | 59.6± 1.1 |
| | PC Aver-*Sample* | 40.0± 1.4 | 66.3± 1.0 | 61.4± 0.5 | 76.4± 0.5 | 83.4± 1.2 | 65.5± 0.9 |
| | **ODiS** | **40.9**± 1.4 | **68.4**± 1.0 | **59.1**± 0.5 | **78.4**± 1.0 | **88.2**± 1.0 | **67.0**± 1.0 |
| **8B** | Nemotron-HQ | 36.2± 1.4 | 62.1± 1.0 | 58.4± 0.5 | 84.3± 1.2 | 74.9± 1.0 | 63.2± 1.0 |
| | PPL-sample | 36.2± 1.4 | 63.1± 1.0 | 57.9± 0.5 | 84.9± 1.1 | 73.3± 1.0 | 63.1± 1.0 |
| | DSIR | 30.9± 1.4 | 49.3± 1.0 | 55.8± 0.5 | 77.8± 1.3 | 71.7± 1.1 | 57.1± 1.0 |
| | PC-Aver-*Top* | 37.1± 1.4 | 59.1± 1.0 | 54.2± 0.5 | 72.4± 1.4 | 75.2± 1.0 | 59.6± 1.1 |
| | PC-Aver-*Sample* | 41.1± 1.4 | 65.7± 1.0 | 59.5± 0.5 | 85.1± 1.1 | 75.5± 1.0 | 65.4± 1.0 |
| | **ODiS** | **40.1**± 1.4 | **67.2**± 1.0 | **57.6**± 0.5 | **85.5**± 1.1 | **77.3**± 1.0 | **65.5**± 1.0 |

Table 3: Performance across methods with model sizes 400M, 1.5B, and 8B and data budgets 100B, 200B, 25B tokens, respectively. The results are reported as percentage accuracy with standard error.

et al., 2019; Touvron et al., 2023). Methods like QuRating (Wettig et al., 2024) and FineWeb-Edu (Penedo et al., 2024) utilize classifiers that focus on specific aspects of LLM capabilities, such as reading comprehension and knowledge acquisition. Moreover, works like Ultra-FineWeb (Wang et al., 2025) and DSIR (Xie et al., 2023) utilize a target dataset to guide the classifier in predicting the quality of the data. Although methods with a single classifier can effectively filter out data with certain desirable features, their reliance on a single evaluation dimension limits data diversity and often leads to imbalanced capabilities in the trained models.

**Multi-classifier-based methods.** More recent works have attempted to incorporate multi-dimensional evaluation during data selection. Compared with single-dimensional methods, the combination of multi-dimensional classifiers can provide a more comprehensive evaluation of data. However, it remains a challenge to balance the influence from different dimensions. One line of the research typically assigns weights to each dimension through performance tests on small proxy models (Zhuang et al., 2025; Bai et al., 2025). However, the dimensional correlations are not well-addressed, resulting in bias in the combined scores and reduced data diversity. Another line of the research design sampling ratios for different domains (Liu et al., 2025) to ensure data diversity. However, the inherent overlapping of the domains will still lead to bias in the selected data. As a result, the traditional top-$k$ method is ineffective in the multi-dimensional setting, leaving the question of integrating multi-dimensional evaluations open.

## 5 CONCLUSION

In this work, we investigated the underlying cause of bias in score-based data selection and identified that neglecting diversity leads to non-monotonicity between the dataset scores and model performance. To address this issue, we proposed ODiS, a method that explicitly mitigates the correlation between different data features while ensuring data diversity and retaining high-quality data. The experiment results demonstrated that ODiS can effectively mitigate inter-dimensional correlation, enhance data diversity, and consistently improve model performance across various downstream tasks compared to several baselines. These findings highlight that effective data selection for pre-training models should consider quality and diversity jointly, and the correlation between different dimensions must be addressed appropriately. Looking forward, we encourage future data selection works to consider the neglected diversity as a cause of performance degradation and adopt appropriate measures to enhance data diversity.

## ETHICS STATEMENT

The authors confirm that this work adheres to the ICLR Code of Ethics. Our research was conducted in accordance with recognized ethical standards, and we have carefully examined the societal, environmental, and potential misuse implications of our contributions.

## REPRODUCIBILITY STATEMENT

The authors have made extensive efforts to ensure the work's Reproducibility, including datasets, evaluation metrics, methodology, models.

The details of the datasets used in this work are all open-sourced, and we have described them in the Section 3.1. The evaluation metrics and prompt for the dimensions can be found in Appendix B and Appendix C. The details of the evaluation benchmarks are described in Section 3.1. The proposed method is described in detail, and the pseudocode is provided. All the implementation details are provided during the description. We utilized the LLaMA-3 model as the base model and adjusted the parameters to obtain a 1.5B-parameter model for training. Besides, the Roberta model can be obtained on the HuggingFace website. All the experimental results are reproducible, and we have averaged the results from multiple experiments to ensure an accurate result.

We believe these detailed descriptions are sufficient to reproduce our results.

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

## A USE OF LARGE LANGUAGE MODELS

During manuscript preparation, we occasionally utilized large language models (LLMs) to refine language expression, such as improving sentence fluency, and enhancing readability. The model was not involved in generating original research contributions, including research direction formulation, methodologies selection, experiment designs, results analysis. All the core intellectual work, such as idea development, experiment execution, and results interpretation, was carried out independently by the authors. Any linguistic suggestions offered by the LLM were carefully reviewed and selectively incorporated, ensuring that accuracy, originality, and scholarly integrity were fully maintained. The authors alone take responsibility for the research content and conclusions, and the LLM is not listed as a contribution or author.

## B 11 EVALUATION DIMENSIONS

To comprehensively evaluate the quality and usefulness of training data, we design an 11-dimensional evaluation framework. These dimensions aim to capture complementary aspects of data that jointly determine its contribution to pretraining large language models (LLMs). Specifically, the dimensions are grouped into four general categories: language quality, which reflects the clarity and fluency of expression; knowledge quality, which measures the depth, diversity, and utility of information; comprehension difficulty, which reflects the complexity of content and its potential to improve generalization; and information quality, which ensures factual correctness and completeness. Together, these dimensions provide a multi-perspective evaluation of data, enabling a balanced assessment of both quality and diversity. Below, we provide the details of each category:

1. **Language quality:** LLMs fundamentally rely on languages to understand the content and interact with humans. High-quality data contributes directly to improving models' comprehension and generation abilities. A high-quality document should present ideas in a logical and rational organization, avoid redundant and irrelevant content, and use accurate spelling and grammar to convey meaning (Penedo et al. (2023)). To capture these properties, we evaluate language quality in three dimensions: (i) *coherence*, which reflects whether the text follows a logical and consistent structure; (ii) *conciseness*, which measures whether the information is conveyed efficiently without unnecessary repetition; and (iii) *spelling/grammar accuracy*, which ensures correctness in word usage and sentence construction.

2. **Knowledge quality:** Beyond linguistic clarity, high-quality data must provide comprehensive knowledge to enrich an LLM's understanding of the world. This dimension mainly measures whether a document contains valuable, diverse, and practical information that can improve the model's reasoning ability and enhance the factual knowledge base (Gunasekar et al. (2023)). Recent work has also shown that the small models can approach the performance of larger ones if trained with reasoning data (Guo et al. (2025)). To comprehensively capture knowledge quality, we define five dimensions: (i) *knowledge depth*, assessing the extent to which a document explores concepts beyond superficial descriptions; (ii) *knowledge richness*, measuring the breadth of covered topics and perspectives; (iii) *reasoning*, capturing the presence of explicit logical inference, argumentation, or step-by-step derivations; (iv) *educational value*, evaluating whether the content provides clear, structured explanations suitable for learning or instruction; and (v) *practical helpfulness*, which assesses the applicability of the knowledge to real-world problems or everyday use.

3. **Comprehension difficulty:** Challenging and complex data can enhance better generalization and adaptability in LLMs. Texts with higher conceptual complexity, specialized domain knowledge, or multi-step expository structures push models to develop deeper comprehension abilities and to generate more professional and domain-specific responses Agrawal & Singh (2023). This dimension therefore evaluates the difficulty of the data, considering factors such as the abstractness of concepts, the level of technicality or professionalism, and the requirement for multi-stage reasoning.

4. **Information Quality:** For models to learn accurate and reliable representations, it is crucial that the training data provides factual, complete, and unambiguous information. Documents with inaccurate or incomplete facts risk propagating errors and degrading downstream performance (Chang et al. (2024)). To evaluate this aspect, we define two evalua-

tion dimensions: (i) *factual accuracy*, which measures the degree to which the document presents correct and verifiable information; and (ii) *completeness*, which reflects whether the information is sufficiently detailed and covers all key aspects of the described topic, thereby reducing the risk of the model learning partial or misleading facts.

## C METRICS AND PROMPT FOR EACH DIMENSION

To evaluate data quality and diversity across multiple dimensions, we design a set of prompts that guide large language models to score documents on 11 distinct dimensions, as described in Appendix B. Each dimension is assessed on a tailored Likert scale ranging from 0 to 3/4/5 points depending on the property being measured, with detailed criteria provided for each score level. The prompts are designed to generate both a quantitative score and a brief qualitative justification, ensuring transparency in the evaluation process. These scores are then aggregated into the score matrix used in our PCA-based data selection framework (see Section 2.3).

```
1  Below is an extract from a web page. Evaluate whether the text
       demonstrates high coherence in terms of language quality. Please
       follow the following guideline to assess the language quality of
       the given extract on a 4 likert scale:
2
3  0 Point: Incomprehensible
4  - The text is grammatically chaotic and difficult to understand.
5  - Severe errors in structure, agreement, and tense prevent
       understanding.
6
7  1 Point: Partially Readable
8  - Some sentences are clear, but overall clarity is lacking.
9  - Noticeable grammatical errors and inconsistency disrupt smooth
       reading.
10
11 2 Points: Moderately Coherent
12 - Occasional language issues but overall understandable.
13 - Logical flow is maintained with some awkward phrasing.
14
15 3 Points: Generally Coherent with Minor Errors
16 - Paragraphs progress logically with minor, infrequent language
       errors.
17 - Sentences are generally well-formed with consistent tense and clear
       subject-verb agreement.
18
19 4 Points: Exceptionally Coherent
20 - The text is grammatically flawless, with precise subject-verb
       agreement and tense usage.
21 - Sentence and paragraph structure is logically ordered and fluid.
22 - Punctuation and syntax enhance the clarity and flow of ideas.
23
24 The extract: {text}
25 After examining the extract:
26 - Briefly justify your total score, up to 50 words.
27 - Conclude begin with the score using the format: "Language
       Coherence Score: <total points>"
```

Listing 1: Prompt for Coherence

```
1  Below is an extract from a web page. Evaluate whether the text
       demonstrates a high level of conciseness. Please follow the
       following guideline to assess the conciseness of the given
       extract on a 4 likert scale:
2
3  0 Point: Excessively Wordy
4  - The extract is filled with redundant, unrelated, or repetitive
       language.
```

```
5   - Nearly every sentence could be significantly shortened or removed
        without loss of meaning.
6   - Core ideas are obscured or lost in verbosity.
7
8   1 Point: Somewhat Wordy
9   - The text is clear but contains noticeable repetition or unnecessary
        words.
10  - Some sentences are overly elaborate.
11
12  2 Points: Moderately Concise
13  - The extract avoids major redundancy but may include some
        unnecessary elaboration.
14  - Most sentences convey meaning efficiently, though small
        improvements in brevity are possible.
15  - The main points are clear and not lost in superfluous language.
16
17  3 Points: Concise and Effective
18  - Ideas are expressed clearly and directly, with minor redundancy or
        unnecessary details.
19  - Minimal to no repetition or fluff.
20
21  4 Points: Exceptionally Concise
22  - Every word is essential and contributes directly to the meaning.
23  - No repetition, filler, or unnecessary elaboration.
24  - The writing is focused, impactful, and efficient.
25
26
27  - The extract: {text}
28
29  After examining the extract:
30   - Briefly justify your total score, up to 50 words.
31   - Conclude begin with the score using the format: "Language
        Conciseness Score: <total points>"
```

Listing 2: Prompt for Conciseness

```
1  Below is an extract from a web page. Evaluate whether the text
       demonstrates high accuracy of word usage, which contributes to
       the as overall language quality. Please follow the following
       guideline to assess the accuracy of word usage in the given
       extract on a 4 likert scale:
2
3  0 Points: Severe Inaccuracy
4  - The extract contains frequent incorrect word usages.
5  - Frequent typos, incorrect word forms, or misuse of words make the
       text almost unreadable.
6  - Errors severely hinder understanding.
7
8  1 Points: Limited Accuracy
9  - Spelling mistakes appear regularly but are not overwhelming.
10 - Occasional misuse of words or minor typos affect clarity.
11 - The overall message is still understandable but occasionally
       unclear.
12
13 2 Points: Moderate Accuracy
14 - Most of the text is correctly spelled, with some minor errors or
       infrequent typos.
15 - Occasional confusion between similar-sounding words may appear but
       does not significantly affect meaning.
16 - The extract remains mostly readable and understandable.
17
18 3 Points: Strong Accuracy
19 - Spelling is generally correct throughout.
20 - Only rare, minor typos or homophone errors are present, and they do
       not interfere with comprehension.
```

```
21   - The extract demonstrates clear attention to written accuracy.
22
23   4 Points: Perfect Accuracy
24   - The extract is free from any spelling errors, typos, or homophone
        confusion.
25   - All words are used appropriately and are correctly spelled.
26   - The writing is polished and precise, reflecting excellent language
        control.
27
28   The extract: {text}
29
30   After examining the extract:
31    - Briefly justify your total score, up to 50 words.
32    - Conclude begin with the score using the format: "Language Spelling
         Accuracy Score: <total points>"
```

Listing 3: Prompt for Spelling Accuracy

```
1   Below is an extract from a web page. Evaluate whether the text
        demonstrates an appropriate depth of knowledge, particularly with
        regard to the grade level it targets. The following gudeline is
        used to assess whether a text has a high knowledge depth on a 5
        likert scale:
2
3   0 Points: No Knowledge Depth
4   - The extract contains no meaningful or accurate knowledge.
5   - It lacks substance entirely and offers no educational value at any
        grade level.
6
7   1 Point: Shallow and Common Knowledge for Pre-K to Grade 1
8   - The content is understandable even to early primary grades (Pre-K
        to Grade 1).
9   - Contain simple, basic facts or common knowledge (e.g., basic facts
        like "grass is green" or "2 + 2 = 4").
10
11  2 Points: Basic Knowledge for Lower Grades (Grades 2-4)
12  - The content is at lower elementary levels.
13  - Introduces simple concepts and provides very short, basic
        explanations.
14  - Requires understanding of simple definitions and explicit
        information.
15
16  3 Points: Introductory Knowledge for Middle Grades (Grades 5-7)
17  - Understandable for upper elementary to early middle school.
18  - Explains foundational concepts with some detail and structure.
19  - Some depth is present. It may require understanding of
        cause-and-effect relationships and ability to follow multi-step
        explanations.
20
21  4 Points: Substantive Knowledge for Secondary Levels (Grades 8-12)
22  - Content is well-developed and appropriate for high school.
23  - Explores concepts in depth, including underlying principles,
        reasoning, and potential implications.
24  - Characterized by complex sentence structures, theoretical concepts,
        evidence or examples to support points; resembles textbook
        content.
25
26  5 Points: Advanced Knowledge Depth (college-level or graduate-level)
27  - The extract reflects college-level or graduate-level understanding.
28  - The knowledge is usually only known to the professional people in a
        certain field.
29  - May presents complex information, including detailed analysis,
        theoretical frameworks, multiple perspectives, and nuanced
        arguments.
30
```

```
31  The extract: {text}
32
33  After examining the extract:
34  - Briefly justify your total score, up to 50 words.
35  - Conclude begin with the score using the format: "Knowledge Depth
       Score: <total points>"
```

Listing 4: Prompt for Knowledge Depth

```
1   Below is an extract from a web page. Evaluate whether the text
       demonstrates a high degree of knowledge density in its content.
       The following curriculum is used to assess whether a text has
       dense knowledge on a 4 likert scale:
2
3   0 Point: No Meaningful Knowledge
4   - The extract lacks any meaningful or specific content.
5   - No concrete facts, data, or identifiable concepts
6
7   1 Point: Minimal Knowledge Content
8   - Contains only 1-2 disjointed factual statements
9   - No context, sourcing, or explanation
10
11  2 Points: Moderately Knowledge Density
12  - The extract includes several points of useful knowledge.
13  - Support with some details, examples, or explanations.
14
15  3 Points: Substantially Rich in Knowledge
16  - The content provides a well-rounded and informative discussion.
17  - Ideas are explained with clarity and supported by relevant details
       or examples.
18
19  4 Points: Exceptionally Knowledge-Rich
20  - The extract offers a dense, nuanced, and well-connected
       presentation of knowledge.
21  - The content shows breadth and depth, encouraging comprehensive
       understanding.
22
23  The extract: {text}
24
25  After examining the extract:
26  - Briefly justify your total score, up to 50 words.
27  - Conclude begin with the score using the format: "Knowledge
       Richness Score: <total points>"
```

Listing 5: Prompt for Knowledge Richness

```
1   Below is an extract from a web page. Evaluate whether the text
       demonstrates a high level of reasoning level. The following
       curriculum is used to assess whether a text has a high reasoning
       level:
2
3   0 Points: No Reasoning Present
4   - The text lacks any evidence of thinking or reasoning from the
       writer.
5
6   1 Point: Minimal Reasoning
7   - Some claims are made, but reasoning is largely absent or extremely
       shallow.
8   - No causal relationships or inferential steps are evident.
9   - Readers are not encouraged to reflect or engage intellectually.
10
11  2 Points: Limited Reasoning
12  The text demonstrates some basic thinking and reasoning, such as:
13  - a straightforward application of a known technique
```

```
14  - simple analysis of a problem.
15
16  3 Points: Moderate Reasoning
17  The text demonstrates adequate level thinking and reasoning, such as
18  - a consideration of multiple approaches to a problem.
19  - A discussion of the trade-offs between different solutions.
20
21  4 Points: Strong Reasoning
22  The text demonstrates significant thinking and reasoning, such as:
23  - Multi-step reasoning chains to solve a complex problem.
24  - Advanced reasoning patterns often used in specialized science
        domains
25
26  5 Points: Exceptional Reasoning Quality
27  The text exemplifies exceptional thinking and reasoning, such as:
28  - A highly innovative and creative approach to solving a complex
        problem in specialized domains.
29  - Combining multiple reasoning and thinking techniques, with novel
        abstraction of the problem.
30
31  The extract: {text}
32
33  After examining the extract:
34   - Briefly justify your total score, up to 50 words.
35   - Conclude begin with the score using the format: "Knowledge
        Reasoning Score: <total points>"
```

Listing 6: Prompt for Reasoning Level

```
1  Below is an extract from a web page. Evaluate whether the page has a
       high educational value for teaching from kindergarten to graduate
       education. The following curriculum is used to assess whether a
       text has a high educational value on a 3 point scale:
2
3  **0 Point: No Educational Value**
4  - Not even a single bit of information is worth learning.
5  - Note that if there is even a single bit of information that is
       worth learning, the score should be at least 1 point.
6
7  **1 Point: Minimal Educational Relevance**
8  - The extract provides some useful information pertinent that is
       worth learning or teaching, but does not align closely with
       educational standards.
9  - It may include a large amount of non-educational content (e.g.,
       advertisements, promotional material) that detracts from its
       usefulness.
10
11  **2 Points: Suitable for Educational Use**
12  - The extract provides a lot of useful information that is worth
       learning or teaching. The content is fluent and coherent.
13  - It may include a small amount of non-educational content. It may
       have limitations, such as incomplete coverage or extraneous
       information.
14
15  **3 Points: Highly Relevant and Beneficial**
16  - The extract has very high educational value. It contains high
       density of information that is worth learning or teaching, either
       for any level of education.
17  - Content is clear, consistent, and focused, with minimal irrelevant
       information.
18  - May resemble a snippet from a textbook, tutorial, exercises,
       solutions, or any structured learning materials.
19
20  The extract:
21  {text}
```

```
22
23  After examining the extract:
24  - Briefly justify your total score, up to 50 words.
25  - Conclude begin with the score using the format: "Educational score:
        <the assigned score>"
```

Listing 7: Prompt for Educational Value

```
1  Below is an extract from a web page. Evaluate whether the content
       demonstrates a high degree of practical helpfulness, particularly
       in terms of offering applicable knowledge for real-world utility.
       The following curriculum is used to assess whether a text has a
       high practical helpfulness on a 4 likert scale:
2
3  0 Points: No Practical Helpfulness
4  - The extract contains no useful or applicable knowledge.
5  - May be purely entertainment or advertisement with zero actionable
       takeaways
6  - May contain misinformation or harmful suggestions
7
8  1 Point: Minor Utility
9  - The text may hint at applicable ideas but lacks clarity,
       specificity, or guidance.
10 - It is too general or abstract to be put into use.
11
12 2 Points: Moderately Helpfulness
13 - The knowledge can be applicable in some uncommon scenarios (targets
       <1% audience) that only relate to a small portion of people.
14
15 3 Points: Broadly Helpful
16 - The extract includes practical information that could be applied in
       common contexts.
17 - Offers validated strategies for common needs
18
19 4 Points: Substantially Helpful
20 - The extract offers clear, applicable knowledge or skills that are
       useful in real-world scenarios that frequently occur.
21 - Addresses frequent pain points (>10% audience)
22
23
24 The extract: {text}
25
26 After examining the extract:
27  - Briefly justify your total score, up to 50 words.
28  - Conclude begin with the score using the format: "Knowledge
        Practical Helpfulness Score: <total points>"
```

Listing 8: Prompt for Practical Helpfulness

```
1  Here is an extract from a webpage. Please evaluate the percentage of
       the global population that is likely to be able to comprehend the
       knowledge text. The following scale is used to assess the
       comprehension difficulty, with a 5-point Likert scale:
2
3  0 Points: No value to understand
4  - The content is incomprehensible due to its low language quality.
5  - Contains gibberish, severe grammar errors, or formatting problems.
6  - Examples: Advertisement, machine-translated nonsense, corrupted text
7
8  1 Point: Universal Comprehension
9  - The content is very simple and direct, easily understood by the
       vast majority of people.
10 - Requies basic vocabulary (<4th grade level), commonsense knowledge,
       with no jargon.
```

```
11  - Examples: Weather reports, simple recipes, basic safety instructions
12
13  2 Points: Majority Effortless
14  The content is clear and easily understandable for almost everyone,
        with only a very small percentage finding it difficult.
15  - Requires conversational language level and general world knowledge
16  - Examples: social media posts, most new articles
17
18  3 Points: Educated Majority
19  - The content is accessible to the majority of people, with some
        difficulty, but most people should be able to understand and
        comprehend it after some effort.
20  - Requires high school reading level and secondary education concepts
21  - Examples: Government pamphlets, workplace training manuals, simple
        financial advice.
22
23  4 Points: Specialized Audience
24  - The content is understood by a small portion of people, but it
        remains challenging for the majority.
25  - The content may require some expertise.
26  - Requires undergraduate-level training in field
27  - Examples: College textbooks, legal contracts, financial advice
28
29  5 Points: Expertise
30  - The content may be very professional or academic.
31  - Requires graduate-level expertise.
32  - Examples: Quantum physics proofs, AI architecture patents, genomic
        research
33
34  Extract:
35   {text}
36
37  After reviewing the text:
38  Briefly justify your total score in up to 50 words.
39  Conclude begin with the score using the format: "Comprehension
        Difficulty Score: "
```

Listing 9: Prompt for Comprehension Difficulty

```
1  Here is an extract from a webpage. Evaluate whether the content
       demonstrates a high level of factual accuracy as part of its
       overall information quality.
2  Note that:
3  - the text may include some facts that are unknown to you. In these
       cases, you can ignore these unknown or uncertain facts and only
       focus on identify those obvious factual errors that are known to
       you.
4  - In some special contexts, such as fictions, it is allowed to
       contain some imaginary facts.
5
6
7  The following guideline is used to assess the factual accuracy, with
       a 3-point Likert scale:
8
9  0 Point: Evidently Inaccurate
10  - The extract is filled with incorrect information.
11  - Key claims are demonstrably wrong or contradict well-established
        facts.
12
13  1 Point: Highly Unreliable
14  - The extract contains multiple factual inaccuracies or distortions.
15  - Misleading phrasing or vague statements obscure the truth.
16  - While not entirely false, it cannot be trusted as a reliable source
        of information.
17
```

```
18  2 Points: Generally Accurate with Minor Issues
19  - <2 minor errors in peripheral details
20  - Occasional imprecise language without distorting meaning
21  - Preserves core truth despite technical imperfections
22
23  3 Points: Accurate and Trustworthy
24  - No detectable errors in verifiable claims.
25
26
27  Extract:
28   {text}
29
30  After reviewing the text:
31  - Briefly justify your total score in up to 50 words.
32  - Conclude begin with the score using the format: "Information
        Factual Accuracy Score:"
```

Listing 10: Prompt for Factual Accuracy

```
1   Here is an extract from a webpage. Evaluate whether the content
        demonstrates a high degree of completeness, specifically in terms
        of how fully the topic is covered and whether the information is
        presented with sufficient context. The following scale is used to
        assess the information completeness, with a 4-point Likert scale:
2
3
4   0 Point: Severely Incomplete
5   The extract offers only fragments of information or vague references
        to the topic.
6   Key background, definitions, or context are missing.
7   The presentation leaves readers with more questions than answers.
8
9   1 Point: Limited Completeness
10  The extract touches on parts of the topic but leaves significant gaps.
11  It may assume prior knowledge or skip necessary context.
12  Information is partial or unevenly distributed.
13
14  2 Points: Moderately Complete
15  The extract introduces the main topic and provides sufficient context
        to follow the discussion.
16  Some areas may be underdeveloped or missing, but overall
        understanding is possible.
17  It resembles a summary or introductory passage.
18
19  3 Points: Substantially Complete
20  The extract covers the topic in a well-rounded and balanced manner.
21  Most relevant aspects are addressed, with clear and sufficient
        context.
22  There may be minor omissions, but they do not disrupt comprehension.
23
24  4 Points: Exceptionally Complete
25  The extract thoroughly explores the topic with comprehensive coverage.
26  All necessary context is included, with no critical gaps.
27  It reflects a deep and well-structured presentation that anticipates
        and answers potential reader questions.
28
29  Extract:
30   {text}
31
32  After reviewing the text:
33  Briefly justify your total score in up to 50 words.
34  Conclude begin with the score using the format: "Information
        Completeness Score: "
```

Listing 11: Prompt for Completeness

## D  MODEL-BASED SCORER

We selected RoBERTa-base as the foundation for our scorer due to its cost-effectiveness. The maximum context window was set to 512 tokens. Each model was finetuned for 5 epochs with a batch size of 256. The learning rate was dynamically adjusted based on validation performance. We trained the RoBERTa-base scorers to regress the FineWeb-Edu 460k samples to the PCA-transformed scores. A 20% set was used for validation. The optimized learning rate and the resulting performance of the RoBERTa scorers are reported as follows:

| Model | Learning Rate | Accuracy (%) | F1 | Spearman Corr |
|-------|--------------|-------------|------|--------------|
| PC1 | 5e-5 | 72.1 | 0.725 | 0.92 |
| PC2 | 3e-5 | 62.7 | 0.611 | 0.68 |
| PC3 | 1e-4 | 63.3 | 0.603 | 0.67 |
| PC4 | 3e-5 | 68.9 | 0.620 | 0.55 |

Table 4: Performance comparison of different models. We report the learning rate, Accuracy, F1 score and spearman correlation coefficient.

## E  COMPUTATIONAL COST ANALYSIS

We analyze the computational cost of our proposed framework, which consists of four stages: labeling the reference dataset, training the Roberta-based scorers, annotating the large-scale dataset, and pre-training the model on the selected data.

**Labeling reference dataset** For the Fineweb-Edu dataset with 460k samples, the average input sequence consists of 750 data tokens and 290 prompt tokens, with an average generation length of 50 tokens. When labeling the reference dataset across 11 dimensions, the cumulative consumption is approximately $5.5 \times 10^9$ tokens.

Following (Kaplan et al., 2020; Hoffmann et al., 2022), we approximate the FLOPs for training a Transformer-based model using Equation 5:

$$C_{\text{train}} \approx 6 \times P \times D_{\text{train}} \times E, \tag{5}$$

where $P$ denotes size of model parameter, $D_{\text{train}}$ denotes the tokens of the training set, and $E$ denotes the number of training epochs. Similarly, the inference FLOPs can be approximated as:

$$C_{\text{infer}} \approx 2 \times P \times D_{\text{infer}}, \tag{6}$$

where $D_{\text{infer}}$ denotes the number of tokens to infer on.

**Training the scorer** We train 4 RoBERTa-based scorers, with 5 epochs each, on the Fineweb-Edu dataset. Therefore, the total cost is about $1.04 \times 10^{18}$ FLOPs.

**Data Annotation.** The trained scorers are used to annotate the entire dataset. This requires inference over the full dataset for each of the 4 scorers. The total cost is about $2.2 \times 10^{21}$ FLOPs.

**Target Model Training.** Finally, we train the target model (with 1.5B parameters) on the selected dataset with 100B token training budget. The training cost is about $9 \times 10^{20}$ FLOPs.

While the annotation cost is comparable to the training budget of the 1.5B model under 100B token budget, we emphasize that this is a one-time fixed investment. Since the annotated data and trained scorers can be reused to filter large training datasets or train significantly larger models (e.g., 7B+), the cost is effectively amortized at scale, making the method highly cost-efficient for real-world pre-training.

## F  BENCHMARKS SELECTION

To select appropriate downstream benchmarks that can effectively reflect the model performance, we observe the accuracy fluctuation as the trained data increases, with results reported in Figure 7. Arc-C, Arc-E, hellaswag, piqa, and sciq have obvious variation as the training progresses, while the rest of the benchmarks have a smaller performance improvement. Since our model and the trained data budget are relatively small, some benchmarks can not obviously reflect the training outcomes of the model. Therefore, we select Arc-C, Arc-E, hellaswag, piqa, and sciq as our benchmarks.

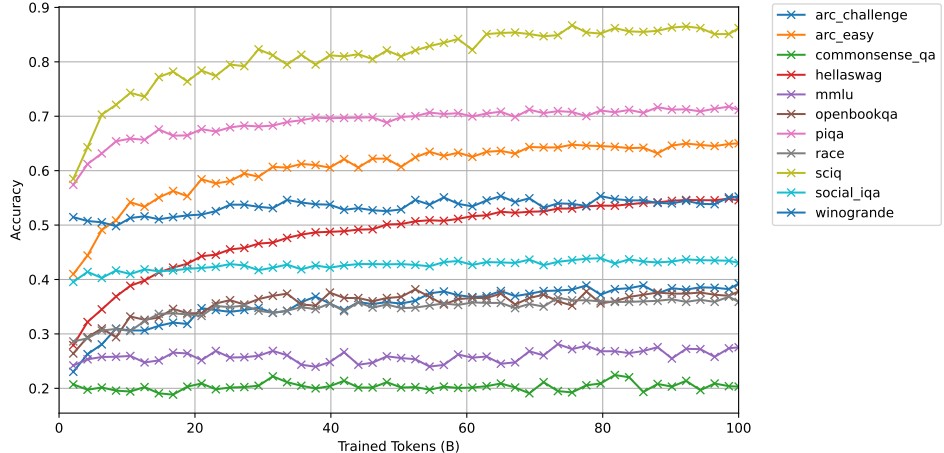

Figure 7: Performance across downstream tasks

## G  SCORE DISTRIBUTION ACROSS DIFFERENT PC

Figure 8 demonstrates the score distribution over different PC dimensions on different domains. The domains are pre-devided by the Nemotron-CC dataset. We can observe that different PC dimensions emphasize distinct aspects, and joint selection across dimensions enhances data diversity.

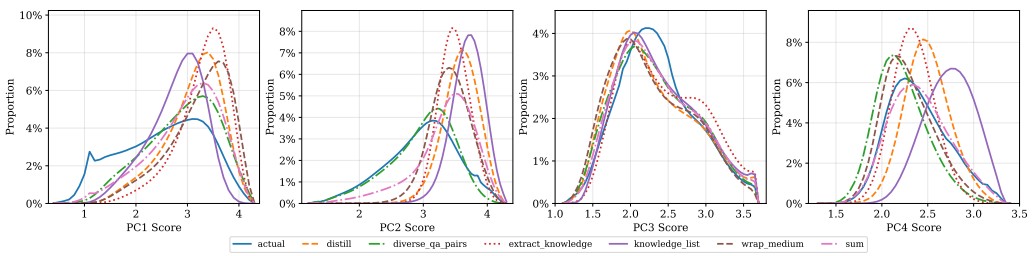

Figure 8: Score distribution over different domains.

## H    RESULTS WITH SINGLE PC

Table 5 demonstrates the results of data selected with the single PC scorer. We can observe that each PC exhibits strength in certain area: PC1 and PC4 perform better on Arc-C and Arc-E, indicating a better ability at general knowledge, while PC2 and PC3 perform better on Hellaswag and PIQA, indicating a better ability for commonsense and physical reasoning. Moreover, models trained with top-scored data from each PC dimension consistently underperform, while sampling from a larger score range enhances the performance. These results highlight that different PC scorers focus on distinct data features and using one of them alone can not achieve the best performance.

| Method | Arc-C | Arc-E | Hellaswag | SCIQ | PIQA | Average |
|---|---|---|---|---|---|---|
| PC1-top | 0.3635 | 0.6035 | 0.4309 | 0.811 | 0.4289 | 0.5705 |
| PC2-top | 0.3072 | 0.5097 | 0.5567 | 0.727 | 0.4483 | 0.5707 |
| PC3-top | 0.3311 | 0.5551 | 0.6178 | 0.741 | 0.4386 | 0.6059 |
| PC4-top | 0.4053 | 0.7041 | 0.4484 | 0.879 | 0.4350 | 0.6245 |
| PC1-Sample | 0.3951 | 0.6519 | 0.5464 | 0.863 | 0.7116 | 0.6336 |
| PC2-Sample | 0.3686 | 0.6103 | 0.5759 | 0.765 | 0.7448 | 0.6129 |
| PC3-Sample | 0.3686 | 0.6557 | 0.6112 | 0.846 | 0.7579 | 0.6479 |
| PC4-Sample | 0.4087 | 0.6860 | 0.5356 | 0.861 | 0.7318 | 0.6446 |

Table 5: Performance across PC dimensions.

