# OpenReview forum: "Learning from the Best, Differently: A Diversity-Driven Rethinking on Data Selection"
_ICLR.cc/2026/Conference — Submitted to ICLR 2026_

### Official Review · Reviewer_wRju · 2025-10-20

**Soundness:** 3
**Presentation:** 3
**Contribution:** 3
**Rating:** 4
**Confidence:** 4

**Summary:**

This paper proposes ODiS (Orthogonal Diversity-Aware Selection), a data selection method for LLM pre-training that addresses the observation that directly selecting top-scored data often degrades downstream performance. Authors identify neglected diversity as the fundamental cause of this non-monotonicity between dataset scores and model performance. Their approach evaluates data across 11 dimensions (covering language quality, knowledge quality, comprehension difficulty, and information quality), applies PCA to decorrelate these dimensions into orthogonal components, trains RoBERTa-based scorers to predict scores along each principal component, and constructs training datasets by selecting top-scored data from each orthogonal dimension. Experiments on a 1.5B parameter model trained with 100B tokens show ODiS outperforms baseline methods on five downstream benchmarks.

**Strengths:**

1. The paper provides valuable analysis showing that top-scored data consistently underperforms compared to sampling from broader score ranges (Figure 1), identifying neglected diversity as the root cause.

2. Using PCA to decorrelate evaluation dimensions is a systematic and well-motivated solution.

3. The paper convincingly demonstrates improved diversity through multiple analyses. UMAP visualizations showing broader distribution (Figure 2a, 4b), increased pairwise distances (Figure 2b, 4a), correlation analysis revealing dimension orthogonality (Figure 5), and minimal inter-dimension overlap (Figure 6b).

**Weaknesses:**

1.  All experiments use only the Nemotron-CC Chinese dataset with a single 1.5B model size and 100B token budget. No experiments on English data, different model scales (e.g., 400M, 7B, as conducted in existing papers on LLM data selection) or varying data budgets are provided. This raises serious questions about whether findings generalize across languages, scales, and domains.

2. The paper mentions several recent multi-dimensional methods (Meta-Rater, QuadMix, SoftDedup) in related work but doesn't compare against them experimentally. The "PC Average" baseline is somewhat weak and weighted combinations of dimensions or other aggregation strategies aren't tested.
I think critical missing baselines include:
- (a) sampling uniformly from each of the 11 original dimensions without PCA.
- (b) simple clustering-based diversity methods.
- (c) domain mixing strategies.

3. The paper doesn't establish that orthogonalization specifically is the key factor vs. simply using multiple diverse metrics. The comparison with "PC Average" (Table 1) shows ODiS performing better, but this doesn't isolate orthogonalization's contribution.

4. The 11 dimensions appear manually designed based on intuition and prior work, without principled justification for why these specific dimensions or this specific decomposition is optimal. Different domains (code, scientific text, dialogue) might require different dimensions.

5. Training 11-dimensional scorers with GPT API is expensive (460k examples in FineWeb-Edu mentioned for correlation analysis), plus training K RoBERTa models. These are missing:
- total computational cost vs baselines
- cost-performance tradeoffs

**Questions:**

1. These analyses are beneficial:
- (a) sensitivity to dimension definitions
- (b) whether learned/discovered dimensions would work better
- (c) how to adapt dimensions for different domains
- (d) whether all 11 dimensions are necessary (some may be redundant even before PCA)

2. Why is PCA specifically the right transformation? Other decorrelation methods aren't considered.

3. While Figure 6b shows 2% overlap after tokenization, this still represents millions of tokens of redundancy at 100B scale. This overlap helps or hurts?

---

> ### Author Response · Authors · 2025-11-23
> **Response to Reviewer wRju (1/n)**
>
> Thank you for your valuable feedback to help us improve our paper. We have revised our paper based on your feedback. We detail our response below and please kindly let us know if our response addresses your concerns.
>
> > (W1) All experiments use only the Nemotron-CC Chinese dataset with a single 1.5B model size and 100B token budget. No experiments on English data, different model scales (e.g., 400M, 7B, as conducted in existing papers on LLM data selection) or varying data budgets are provided. This raises serious questions about whether findings generalize across languages, scales, and domains.
>
> We thank the reviewer for the constructive feedback on experiment settings. We address the concerns on dataset language, model scale, and generality as follows.
>
> First, we sincerely apologize for the error in the original manuscript that describing Nemotron-CC as a Chinese dataset. We clarify that Nemotron-CC is a large-scale English dataset. Thus, our experiments are conducted on a standard English benchmark, aligning with the majority of data selection literature. We have corrected this description in the revision.
>
> To address concerns about model scaling, we have conducted an additional experiment with a model with 400M parameters. The results are presented below:
>
> | Model | Method | Arc-C | Arc-E | Hellaswag | SCIQ | PIQA | Average |
> | :--- | :--- | :---: | :---: | :---: | :---: | :---: | :---: |
> | **1.5B** | Random | 35.0 ± 1.4 | 62.7 ± 1.0 | 58.3 ± 0.5 | 85.5 ± 1.1 | 74.5 ± 1.0 | 63.2 ± 1.0 |
> | | Nemotron-HQ | 37.3 ± 1.4 | 64.6 ± 1.0 | 57.7 ± 0.5 | 83.9 ± 1.1 | 73.6 ± 1.0 | 63.4 ± 1.0 |
> | | PPL-*Top* | 37.9 ± 1.4 | 62.8 ± 1.0 | 54.7 ± 0.5 | 83.4 ± 1.2 | 74.7 ± 1.0 | 62.7 ± 1.0 |
> | | PPL-*Sample* | 36.1 ± 1.4 | 64.3 ± 1.0 | 58.4 ± 0.5 | 85.8 ± 1.1 | 74.8 ± 1.0 | 63.9 ± 1.0 |
> | | DSIR | 27.8 ± 1.3 | 48.5 ± 1.0 | 54.6 ± 0.5 | 78.5 ± 1.3 | 71.0 ± 1.1 | 56.1 ± 1.0 |
> | | PC Aver-*Top* | 35.3 ± 1.4 | 57.9 ± 1.0 | 53.3 ± 0.5 | 71.9 ± 1.4 | 74.5 ± 1.0 | 58.6 ± 1.1 |
> | | PC Aver-*Sample* | 39.2 ± 1.4 | 64.8 ± 1.0 | 58.9 ± 0.5 | 80.6 ± 1.3 | 75.1 ± 1.0 | 63.7 ± 1.0 |
> | | **ODiS** | **41.6 ± 1.4** | **66.9 ± 1.0** | **58.4 ± 0.5** | **85.6 ± 1.0** | **77.4 ± 1.0** | **66.0 ± 1.0** |
> | **400M** | Random | 26.5 ± 1.3 | 49.5 ± 1.0 | 40.6 ± 0.5 | 74.7 ± 1.4 | 67.5 ± 1.1 | 51.7 ± 1.1 |
> | | Nemotron-HQ | 29.6 ± 1.3 | 54.3 ± 1.0 | 40.7 ± 0.5 | 75.4 ± 1.4 | 67.9 ± 1.1 | 53.6 ± 1.1 |
> | | PPL-*Top* | 27.7 ± 1.3 | 50.3 ± 1.0 | 40.0 ± 0.5 | 73.0 ± 1.4 | 67.9 ± 1.1 | 51.8 ± 1.1 |
> | | PPL-*sample* | 28.2 ± 1.3 | 50.7 ± 1.0 | 41.7 ± 0.5 | 76.3 ± 1.4 | 69.9 ± 1.1 | 53.4 ± 1.1 |
> | | DSIR | 24.6 ± 1.3 | 40.7 ± 1.0 | 36.5 ± 0.5 | 66.6 ± 1.5 | 65.0 ± 1.1 | 46.7 ± 1.1 |
> | | PC-Aver-*Top* | 29.2 ± 1.3 | 48.7 ± 1.0 | 41.6 ± 0.5 | 69.9 ± 1.5 | 70.5 ± 1.1 | 52.0 ± 1.1 |
> | | PC-Aver-*Sample* | 30.1 ± 1.3 | 53.2 ± 1.0 | 43.2 ± 0.5 | 75.0 ± 1.4 | 70.1 ± 1.1 | 54.3 ± 1.1 |
> | | **ODiS** | **30.0 ± 1.3** | **54.3 ± 1.0** | **42.5 ± 0.5** | **74.5 ± 1.4** | **72.5 ± 1.0** | **54.8 ± 1.1** |
>
> As shown in the table, ODiS consistently outperforms baselines on the 400M scale. This consistency across two orders of magnitude (400M and 1.5B) demonstrates the robustness of our method and suggests that the benefits follow neural scaling laws. Regarding 7B models, we respectfully note that validating pre-training data selection on models smaller than 3B aligns with the existing works (1B in DoReMi [1], 1B in MATES [2] , 3B in Meta-rater [3]). Since ODiS shows a consistent performance lead at both 400M and 1.5B, we believe these results provide sufficient evidence of its efficacy. Furthermore, we are conducting experiments with a 200B token data budget to validate the performance. The results will be reported soon.
>
> [1] Sang Michael Xie, Hieu Pham, Xuanyi Dong, Nan Du, Hanxiao Liu, Yifeng Lu, Percy S Liang, Quoc V Le, Tengyu Ma, and Adams Wei Yu. Doremi: Optimizing data mixtures speeds up language model pretraining. Advances in Neural Information Processing Systems, 36:69798–69818, 2023a.
>
> [2] Zichun Yu, Spandan Das, and Chenyan Xiong. Mates: Model-aware data selection for efficient pretraining with data influence models. Advances in Neural Information Processing Systems, 37: 108735–108759, 2024
>
> [3] Xinlin Zhuang, Jiahui Peng, Ren Ma, Yinfan Wang, Tianyi Bai, Xingjian Wei, Qiu Jiantao, Chi Zhang, Ying Qian, and Conghui He. Meta-rater: A multi-dimensional data selection method for pre-training language models. In Proceedings of the 63rd Annual Meeting of the Association for Computational Linguistics (Volume 1: Long Papers), pp. 10856–10896, 2025

---

> ### Author Response · Authors · 2025-11-23
> **Response to Reviewer (2/n)**
>
> > (W2) The paper mentions several recent multi-dimensional methods (Meta-Rater, QuadMix, SoftDedup) in related work but doesn't compare against them experimentally. The "PC Average" baseline is somewhat weak and weighted combinations of dimensions or other aggregation strategies aren't tested. I think critical missing baselines include:  (a) sampling uniformly from each of the 11 original dimensions without PCA, (b) simple clustering-based diversity methods, (c) domain mixing strategies.
>
> We thank the reviewer for the advice. We are currently implementing the above three baselines, and the results will be updated soon.
>
> > (W3) The paper doesn't establish that orthogonalization specifically is the key factor vs. simply using multiple diverse metrics. The comparison with "PC Average" (Table 1) shows ODiS performing better, but this doesn't isolate orthogonalization's contribution.
>
> The orthogonalization is the key factor. The PC average is equivalent to putting all the metrics together with a weighted sum. In contrast, our method selects data from each dimension, without the influence of others. Therefore, we conclude that selecting data from each orthogonalization dimension is key to achieving performance gains.
>
> > (W4) The 11 dimensions appear manually designed based on intuition and prior work, without principled justification for why these specific dimensions or this specific decomposition is optimal. Different domains (code, scientific text, dialogue) might require different dimensions.
>
>
> We thank the reviewer for the concern regarding the design of our metrics. We argue that the specific choice of dimensions is grounded in literature consensus and that our PCA mechanism makes the method robust to the selection of metrics.
>
> The 11 dimensions were not chosen arbitrarily but were synthesized from established dimensions in recent data selection literature. They cover the fundamental features of text quality required for general pre-training: Language quality, Knowledge quality, Comprehension difficulty, and Information Quality (We have explained the details of metrics selection in Section 2.3).
>
> Regarding the optimality of the evaluation metrics, we would like to claim that ODiS does not strictly require the input dimensions to be optimal or perfectly independent, because PCA will further process the raw score. Even if our manual selection includes overlapping dimensions, PCA automatically fuses them into a single principal component based on their covariance. As long as the initial dimensions provide sufficient coverage of the quality space, PCA will find the optimal orthogonal basis for selection.
>
> While we designed these 11 dimensions for general-purpose pre-training, ODiS is inherently an extensible framework. The dimensions we chose are domain-agnostic, while extra metrics for specialized domains can be directly plugged in the framework.

---

> ### Author Response · Authors · 2025-11-23
> **Response to Reviewer (3/n)**
>
> > (W5) Training 11-dimensional scorers with GPT API is expensive (460k examples in FineWeb-Edu mentioned for correlation analysis), plus training K RoBERTa models. These are missing: total computational cost vs baselines; cost-performance tradeoffs.
>
> We thank the reviewer for raising the concern about the computational cost, and we would like to address it from following perspectives.
>
> The total training cost is reported with a breakdown in Appendix E. As for the GPT-labeling, the token cost is about $5.5\times10^9$ tokens. The cost is relatively small compared to the resources used in model pretraining. Moreover, this annotation is a one-time investment. Once the 11-dimensional scorers are trained on this seed dataset, they can be applied to filter datasets of any size (e.g., 100B, 1T, or 10T tokens) using only the low-cost inference of RoBERTa models.
>
> The cost comparison between the baselines are as follows. We did not include DSIR in the FLOPs comparison because it is a CPU-based algorithm. However, despite its low computational overhead, its downstream performance is significantly inferior to ODiS and other baselines, indicating that simple heuristics are insufficient for high-quality data selection. Regarding the Nemotron-CC HQ baseline, we note that the cost listed in the table only accounts for the model-based filtering step. However, it relies heavily on synthetic data generation, which constitutes the dominant computational burden. In comparison, the PPL method has a smaller computational cost with a much worse performance. Therefore, when evaluating the cost-performance trade-off, we think ODiS achieves the optimal balance.
>
> | Method | Cost (FLOPs) |
> | :--- | :---: |
> | PPL | $3.63 \times 10^{20}$ |
> | Nemotron-CC HQ | $9.59 \times 10^{20}$ |
> | ODiS | $2.2 \times 10^{21}$ |
>
> **Table:** Computational cost comparison of different methods.
>
> We have conducted the cost-performance trade-off, with different numbers of RoBERTa scorers. As shown in the table, the performance gain becomes marginal when adding more PC dimensions. Using just the first 4 PCs captures about 90% of the performance gain while reducing the inference cost by 30%. This demonstrates that the proposed ODiS is flexible, and the users can select the number of dimensions based on the need.
>
> | Method | Cost ($\times 10^{21}$ FLOPs) | Average (%) |
> | :--- | :---: | :---: |
> | PC1 | 5.5 | 57.1 |
> | PC1-2 | 11 | 61.7 |
> | PC1-3 | 16.5 | 62.6 |
> | PC1-4 | 22 | 65.6 |
> | PC1-5 | 27.5 | 65.6 |
> | PC1-6 | 33 | 66.5 |
>
> **Table:** Performance-cost trade-off analysis.

---

> ### Author Response · Authors · 2025-11-23
> **Response to Reviewer wRju (4/n)**
>
> > (Q1) These analyses are beneficial:
> > (Q1.a) sensitivity to dimension definitions.
>
> Our method is robust to minor variations in dimension definitions. The dimensions are designed to capture all the capabilities that are desired for the model performance. Therefore, even if the definition changes, as long as the feature coverage remains the same, we can remove the redundancy through PCA.
>
> > (Q1.b) whether learned/discovered dimensions would work better.
>
> We chose predefined dimensions based on literature consensus (as described in Section 2.3). Since unsupervised learning (e.g., clustering embeddings) tends to discover semantic topics rather than quality distinctions, the prompt-based annotation is to ensure the scorer focuses on quality. We use human-defined rules to guide the GPT to separate quality features specifically, which are then orthogonalized by PCA.
>
> > (Q1.c) how to adapt dimensions for different domains;
>
> The proposed ODiS is a domain-agnostic framework, which can adapt to specific domains through simple adjustment on the dimension definition. Once the new domain-specific metrics are defined, the downstream pipeline (GPT labeling, PCA, and RoBERTa training) remains unchanged. This flexibility is a key advantage of our approach.
>
> > (Q1.d) whether all 11 dimensions are necessary (some may be redundant even before PCA).
>
> We agree that the raw 11 dimensions contain redundancy. This is exactly why we employ PCA. The 11-dimension intends to cover the capabilities we hope the model will achieve, and PCA is then used to remove inherent redundancy.
>
> > (Q2) Why is PCA specifically the right transformation? Other decorrelation methods aren't considered.
>
> As shown in the case study in Section 2.1 and the correlation analysis in Section 3.3.2, the original dimensions are highly coupled. Consequently, selecting data based on these raw dimensions lacks diversity and leads to performance degradation. To address this, we need to construct orthogonal dimensions to evaluate data, where PCA is a choice to break this coupling. By selecting from orthogonal dimensions, we ensure that the selected data contributes unique, non-overlapping features, maximizing the diversity and quality of the final dataset.
>
> Besides the direct motivation, we choose PCA for transformation for the reasons of ranked variance and parameter-free robustness. Unlike other decorrelation methods like Independent Component Analysis (ICA), PCA orders the orthogonal transformed dimensions by variance. This allows us to capture the most significant parts in data quality (e.g., PC1, PC2) versus those that likely capture noise induced by the annotation process (e.g., PC11). During data selection, we will prioritize the top-ranked dimensions to select the data. Moreover, using non-linear methods (e.g., $\beta$-Variational Autoencoders) would introduce unnecessary complexity and hyperparameters, with unordered dimensions. In contrast, PCA is a non-parametric, closed-form solution, which ensures reproducibility and avoids tuning sensitivity.
>
> While we acknowledge that exploring more advanced transformations is a promising direction for future work, PCA remains the most theoretically grounded and practical starting point for data selection.
>
> > (Q3) While Figure 6b shows 2% overlap after tokenization, this still represents millions of tokens of redundancy at 100B scale. This overlap helps or hurts?
>
> We have tested the results with the deduplication and without the deduplication. The results are reported in the following table, where ODiS-direct is the method that selects tokens from dimensions independently and then merge them together, while ODiS-dedup selects the tokens through the joint score threshold and avoids data repetition. We found that with repeated tokens, the performance endures slight degradation. In the algorithm description, we also indicates that the data should be selected jointly, which avoids the redundancy.
>
> | Method | Arc-C | Arc-E | Hellaswag | SCIQ | PIQA | Average |
> | :--- | :---: | :---: | :---: | :---: | :---: | :---: |
> | ODiS-Direct | 0.394 | 0.656 | 0.581 | 0.846 | 0.771 | 0.650 |
> | ODiS-Dedup | 0.416 | 0.669 | 0.584 |  0.856 | 0.774 |0.660 |

---

> > ### Comment · Reviewer_wRju · 2025-11-24
> >
> > Thanks for your detailed reply. After reviewing your results and analyses for W2, I would consider raising my score.

---

> ### Author Response · Authors · 2025-12-04
> **Response to Reviewer wRju (5/n)**
>
> > (W1) All experiments use only the Nemotron-CC Chinese dataset with a single 1.5B model size and 100B token budget. No experiments on English data, different model scales (e.g., 400M, 7B, as conducted in existing papers on LLM data selection) or varying data budgets are provided. This raises serious questions about whether findings generalize across languages, scales, and domains.
>
> To further response to (W1), we have added experiments with various settings, with model sizes 400M, 1.5B, and 8B under data budgets 100B, 200B, and 25B, respectively. These results are provided in Section 3.3.5, and they validate that the proposed ODiS can effectively scale to different model sizes and data budgets.
>
> | Model | Method | Arc-C | Arc-E | Hellaswag | SCIQ | PIQA | Average |
> | :--- | :--- | :---: | :---: | :---: | :---: | :---: | :---: |
> | **400M** | Nemotron-HQ | 29.6 ± 1.3 | 54.3 ± 1.0 | 40.7 ± 0.5 | 75.4 ± 1.4 | 67.9 ± 1.1 | 53.6 ± 1.1 |
> | | PPL-*sample* | 28.2 ± 1.3 | 50.7 ± 1.0 | 41.7 ± 0.5 | 76.3 ± 1.4 | 69.9 ± 1.1 | 53.4 ± 1.1 |
> | | DSIR | 24.6 ± 1.3 | 40.7 ± 1.0 | 36.5 ± 0.5 | 66.6 ± 1.5 | 65.0 ± 1.1 | 46.7 ± 1.1 |
> | | PC-Aver-*Top* | 29.2 ± 1.3 | 48.7 ± 1.0 | 41.6 ± 0.5 | 69.9 ± 1.5 | 70.5 ± 1.1 | 52.0 ± 1.1 |
> | | PC-Aver-*Sample* | 30.1 ± 1.3 | 53.2 ± 1.0 | 43.2 ± 0.5 | 75.0 ± 1.4 | 70.1 ± 1.1 | 54.3 ± 1.1 |
> | | **ODiS** | **30.0 ± 1.3** | **54.3 ± 1.0** | **42.5 ± 0.5** | **74.5 ± 1.4** | **72.5 ± 1.0** | **54.8 ± 1.1** |
> | **1.5B** | Nemotron-HQ | 40.3 ± 1.4 | 67.9 ± 1.0 | 59.9 ± 0.5 | 74.5 ± 1.0 | 87.1 ± 1.1 | 65.9 ± 1.0 |
> | | PPL-Top | 37.4 ± 1.4 | 63.9 ± 1.0 | 56.4 ± 0.5 | 74.2 ± 1.0 | 84.5 ± 1.2 | 63.3 ± 1.0 |
> | | PPL-*Sample* | 38.2 ± 1.4 | 67.5 ± 1.0 | 60.5 ± 0.5 | 75.6 ± 1.0 | 88.8 ± 1.0 | 66.1 ± 1.0 |
> | | DSIR | 30.2 ± 1.3 | 50.8 ± 1.0 | 58.1 ± 0.5 | 72.2 ± 1.1 | 79.4 ± 1.3 | 58.1 ± 1.0 |
> | | PC Aver-*Top* | 36.6 ± 1.4 | 59.1 ± 1.0 | 54.8 ± 0.5 | 74.9 ± 1.0 | 72.8 ± 1.4 | 59.6 ± 1.1 |
> | | PC Aver-*Sample* | 40.0 ± 1.4 | 66.3 ± 1.0 | 61.4 ± 0.5 | 76.4 ± 0.5 | 83.4 ± 1.2 | 65.5 ± 0.9 |
> | | **ODiS** | **40.9 ± 1.4** | **68.4 ± 1.0** | **59.1 ± 0.5** | **78.4 ± 1.0** | **88.2 ± 1.0** | **67.0 ± 1.0** |
> | **8B** | Nemotron-HQ | 36.2 ± 1.4 | 62.1 ± 1.0 | 58.4 ± 0.5 | 84.3 ± 1.2 | 74.9 ± 1.0 | 63.2 ± 1.0 |
> | | PPL-sample | 36.2 ± 1.4 | 63.1 ± 1.0 | 57.9 ± 0.5 | 84.9 ± 1.1 | 73.3 ± 1.0 | 63.1 ± 1.0 |
> | | DSIR | 30.9 ± 1.4 | 49.3 ± 1.0 | 55.8 ± 0.5 | 77.8 ± 1.3 | 71.7 ± 1.1 | 57.1 ± 1.0 |
> | | PC-Aver-*Top* | 37.1 ± 1.4 | 59.1 ± 1.0 | 54.2 ± 0.5 | 72.4 ± 1.4 | 75.2 ± 1.0 | 59.6 ± 1.1 |
> | | PC-Aver-*Sample* | 41.1 ± 1.4 | 65.7 ± 1.0 | 59.5 ± 0.5 | 85.1 ± 1.1 | 75.5 ± 1.0 | 65.4 ± 1.0 |
> | | **ODiS** | **40.1 ± 1.4** | **67.2 ± 1.0** | **57.6 ± 0.5** | **85.5 ± 1.1** | **77.3 ± 1.0** | **65.5 ± 1.0** |
>
> > (W2) The paper mentions several recent multi-dimensional methods (Meta-Rater, QuadMix, SoftDedup) in related work but doesn't compare against them experimentally. The "PC Average" baseline is somewhat weak and weighted combinations of dimensions or other aggregation strategies aren't tested. I think critical missing baselines include: (a) sampling uniformly from each of the 11 original dimensions without PCA, (b) simple clustering-based diversity methods, (c) domain mixing strategies.
>
> We have incorporated the three requested baselines: (a) Raw-Metric Selection (selecting from original 11 dimensions without PCA), (b) SemDedup (as the clustering-based diversity baseline), and (c) Web-Organizer (as the domain mixing strategy). The results are provided in the Section 3.2 of the revised manuscript. ODiS consistently outperforms the baselines, which indicates the significance of the orthogonal dimension construction and selecting data from each dimension separately.
>
> | Method | Arc-C | Arc-E | Hellaswag | SCIQ | PIQA | Average |
> | :--- | :---: | :---: | :---: | :---: | :---: | :---: |
> | Random | 35.0 ± 1.4 | 62.7 ± 1.0 | 58.3 ± 0.5 | 85.5 ± 1.1 | 74.5 ± 1.0 | 63.2 ± 1.0 |
> | Nemotron-HQ | 37.3 ± 1.4 | 64.6 ± 1.0 | 57.7 ± 0.5 | 83.9 ± 1.1 | 73.6 ± 1.0 | 63.4 ± 1.0 |
> | PPL-*Top* | 37.9 ± 1.4 | 62.8 ± 1.0 | 54.7 ± 0.5 | 83.4 ± 1.2 | 74.7 ± 1.0 | 62.7 ± 1.0 |
> | PPL-*Sample* | 36.1 ± 1.4 | 64.3 ± 1.0 | 58.4 ± 0.5 | 85.8 ± 1.1 | 74.8 ± 1.0 | 63.9 ± 1.0 |
> | DSIR | 27.8 ± 1.3 | 48.5 ± 1.0 | 54.6 ± 0.5 | 78.5 ± 1.3 | 71.0 ± 1.1 | 56.1 ± 1.0 |
> | PC Aver-*Top* | 35.3 ± 1.4 | 57.9 ± 1.0 | 53.3 ± 0.5 | 71.9 ± 1.4 | 74.5 ± 1.0 | 58.6 ± 1.1 |
> | PC Aver-*Sample* | 39.2 ± 1.4 | 64.8 ± 1.0 | 58.9 ± 0.5 | 80.6 ± 1.3 | 75.1 ± 1.0 | 63.7 ± 1.0 |
> | Web-Org *Format* | 36.4 ± 1.4 | 64.5 ± 1.0 | 57.5 ± 0.5 | 84.6 ± 1.1 | 73.7 ± 1.0 | 63.3 ± 1.0 |
> | Web-Org *Topic* | 36.2 ± 1.4 | 62.1 ± 1.0 | 60.3 ± 0.5 | 83.7 ± 1.2 | 75.8 ± 1.0 | 63.6 ± 1.0 |
> | Ori-11-Dim | 41.0 ± 1.4 | 66.5 ± 1.0 | 54.8 ± 0.5 | 86.5 ± 1.1 | 73.4 ± 1.0 | 64.5 ± 1.0 |
> | Semdedup | 39.6 ± 1.4 | 66.4 ± 1.0 | 57.8 ± 0.5 | 85.0 ± 1.1 | 74.4 ± 1.0 | 64.6 ± 1.0 |
> | **ODiS** | **41.6 ± 1.4** | **66.9 ± 1.0** | **58.4 ± 0.5** | **85.6 ± 1.0** | **77.4 ± 1.0** | **66.0 ± 1.0** |

---

### Official Review · Reviewer_KEDT · 2025-10-26

**Soundness:** 3
**Presentation:** 2
**Contribution:** 3
**Rating:** 4
**Confidence:** 3

**Summary:**

This paper concerns the data selection problem in LLM pre-training. The main contribution is not yet another heuristic criterion for assessing corpus “quality,” but rather an insightful argument that correlations between these criteria hinder effective selection. Inspired by this, this work transforms these scores into an orthogonal space (using PCA) and selecting data with top scores in each principal component dimension and thus improves downstream task performance.

**Strengths:**

The logic is clear, and most arguments are supported by convincing empirical evidence:

- There are correlations between existing data selection metrics → Figure 5(a)
- PCA can remove these correlations → Figure 6(b)
- Selecting data according to the PC dimension scores (the first four in practice) improves performance → Table 1
- In contrast, selecting according to only one PC dimension compromises performance → Figure 3(a)

**Weaknesses:**

There are some typos or mistakes in the figures/tables:

- (W1) In Figure 1, the Arc-Easy results are lower than those of the more challenging Arc-Challenge.
- (W2) In Table 1, the results are reported as averages over five domains. However, the average scores appear to be incorrect. This applies to the proposed method (ODiS) as well as to the baselines PC Average-Sample and PC Average-Top, while the averages for DSIR and Random Selection seem correct.

Additionally, (W3) the performance of the RoBERTa-based scorer on the validation set is missing.

**Questions:**

I don’t have any substantive questions at this time.

However, due to potential errors in the report results (see weaknesses), I am unable to provide a clear recommendation for now. I will update my score after the necessary corrections, justifications, and explanations are provided.

---

> ### Author Response · Authors · 2025-11-23
> **Response to Reviewer KEDT (1/n)**
>
> Thank you for your valuable feedback to help us improve our paper. We have revised our paper based on your feedback. We detail our response below and please kindly let us know if our response addresses your concerns.
>
> > (W1) In Figure 1, the Arc-Easy results are lower than those of the more challenging Arc-Challenge.
>
> Thanks for pointing out the mistake. We have mistakenly swapped the figures for the two metrics. We have corrected it in the revised version.
>
> > (W2) In Table 1, the results are reported as averages over five domains. However, the average scores appear to be incorrect. This applies to the proposed method (ODiS) as well as to the baselines PC Average-Sample and PC Average-Top, while the averages for DSIR and Random Selection seem correct.
>
> Thanks for pointing out the mistake. We have mistakenly mixed up some performance results. The results have been updated in the revised version.
>
> > (W3) the performance of the RoBERTa-based scorer on the validation set is missing.
>
> The performance details have been updated in Appendix D. We also provide the performance below for your convenience. The scorer achieves an approximate accuracy of 70% and an F1 score of 0.65. Furthermore, the spearman correlations range from 0.55 to 0.90 across different PCs, with an average correlation of 0.7.
>
> | Model | Learning Rate | Accuracy (%) | F1 | Spearman Corr |
> | :--- | :---: | :---: | :---: | :---: |
> | PC1 | 5e-5 | 72.1 | 0.725 | 0.92 |
> | PC2 | 3e-5 | 62.7 | 0.611 | 0.68 |
> | PC3 | 1e-4 | 63.3 | 0.603 | 0.67 |
> | PC4 | 3e-5 | 68.9 | 0.620 | 0.55 |

---

> > ### Comment · Reviewer_KEDT · 2025-11-27
> >
> > I appreciate the corrections to the typos and the added evidence in your revision. These improvements strengthen the work, and I have raised my score from 4 to 6.

---

> > > ### Author Response · Authors · 2025-11-27
> > > **Response to Reviewer KEDT**
> > >
> > > Thanks for your time and effort in reviewing our paper and rebuttal. We are glad that our response addressed your concerns and greatly appreciate your decision to raise the score. Your constructive feedbacks have enhanced the quality of our paper.

---

### Official Review · Reviewer_nhUc · 2025-10-31

**Soundness:** 2
**Presentation:** 3
**Contribution:** 2
**Rating:** 2
**Confidence:** 4

**Summary:**

This paper identifies a limitation of score-based data selection for LLM pretraining. The authors propose ODiS (Orthogonal Diversity-Aware Selection), which evaluates data across quality-related metrics, applies PCA to decorrelate the dimensions, and trains RoBERTa regressors to predict principal-component scores on large corpora. High-scoring data along each PC dimension is selected to preserve both quality and diversity. Experiments show consistent improvements across multiple benchmarks, outperforming baselines. Additional analyses show improved diversity and reduced inter-dimension redundancy.

**Strengths:**

The proposed ODiS approach is conceptually simple and effective, showing that removing correlation among quality metrics is a useful insight and appropriately implemented via PCA. The experimental results demonstrate consistent gains over widely used baselines. The method seems to be model-agnostic, scalable, and practical. The study highlights a non-monotonic relationship between data quality scores and downstream performance, providing a compelling explanation grounded in diversity.

**Weaknesses:**

The method relies heavily on GPT-based scoring to obtain 11-dimensional metrics. This is a substantial concern, as it introduces bias and cost concerns.

The explanation of how thresholds per PC are chosen is underspecified.

Interpretation of principal components remains unclear, limiting insight into what semantic attributes each dimension captures.

Experiments are limited to a single language and corpus (Chinese Nemotron-CC) and one model scale (1.5B), raising concerns about generality. More evaluation on larger models or multilingual datasets would strengthen conclusions.

**Questions:**

LLM Scoring Bias: Since GPT models produce the initial 11-dimensional scores, how robust is ODiS to inaccuracies or systematic biases in these annotations? Have you experimented with weak or noisy scoring sources?

Generalization: Do improvements persist for larger model sizes or multilingual corpora?

---

> ### Author Response · Authors · 2025-11-23
> **Response to Reviewer nhUc (1/n)**
>
> Thank you for your valuable feedback to help us improve our paper. We have revised our paper based on your feedback. We detail our response below and please kindly let us know if our response addresses your concerns.
>
> > (W1) The method relies heavily on GPT-based scoring to obtain 11-dimensional metrics. This is a substantial concern, as it introduces bias and cost concerns.
>
> We thank the reviewer for raising the concern about the cost and bias, and we would like to clarify these issues as follows.
>
> The total cost for scoring 11-dimension metrics with GPT is about $5.5\times10^9$ tokens. Since we only apply GPT to annotate a small reference dataset and then train lightweight model-based scorers for large-scale annotation, the cost during the GPT-annotation is small compared with subsequent dataset labeling and model pretraining. Besides, using a large model to annotate only a small subset and then training lightweight model-based scorers for large-scale labeling is already a standard practice in pretraining data construction (e.g., Meta-rater [1], Nemotron-CC [2], Phi-1 [3], Llama-3 [4], FineWeb-Edu [5]). Therefore, we believe our approach is both widely applicable and practically feasible for research and industrial deployment.
>
> We agree that LLM-based scorers (e.g., GPT) exhibit certain inherent biases (e.g., verbosity bias or specific style preference). However, we would like to first clarify that LLM judges are competitive with human experts. Existing works like G-Eval [6] and Mt-bench [7] have demonstrated that LLM with specific metrics and prompts (e.g., coherence, consistency) yields results with high correlation to humans. In addition, recent massive-scale datasets like FineWeb [5] and Nemotron-CC [2] explicitly use LLMs for scored-based web data filtering. Following this paradigm, the evaluation results through GPT can well reflect the features of the data.
>
> Moreover, the proposed method can mitigate the systematic bias introduced by GPT-based annotation. Traditional methods that rely on a single score (e.g., an average score or a direct GPT quality score) are vulnerable to the systematic bias from the scorer. In our analysis, we found that there exist correlations between different original evaluation metrics (as described in Section 3.3.2). ODiS addresses this through PCA-based dimensional decoupling. By selecting data from orthogonal dimensions, it ensures the retention of data that excels in specific, independent qualities (e.g., fluency and knowledge depth) even if it scores poorly on certain dimensions.
>
> [1] Xinlin Zhuang, Jiahui Peng, Ren Ma, Yinfan Wang, Tianyi Bai, Xingjian Wei, Qiu Jiantao, Chi Zhang, Ying Qian, and Conghui He. Meta-rater: A multi-dimensional data selection method for pre-training language models. In Proceedings of the 63rd Annual Meeting of the Association for Computational Linguistics (Volume 1: Long Papers), pp. 10856–10896, 2025
>
> [2] Dan Su, Kezhi Kong, Ying Lin, Joseph Jennings, Brandon Norick, Markus Kliegl, Mostofa Patwary, Mohammad Shoeybi, and Bryan Catanzaro. Nemotron-cc: Transforming common crawl into a refined long-horizon pretraining dataset. arXiv preprint arXiv:2412.02595, 2024.
>
> [3] Suriya Gunasekar, Yi Zhang, Jyoti Aneja, Caio C´esar Teodoro Mendes, Allie Del Giorno, Sivakanth Gopi, Mojan Javaheripi, Piero Kauffmann, Gustavo de Rosa, Olli Saarikivi, et al. Textbooks are all you need. arXiv preprint arXiv:2306.11644, 2023.
>
> [4] Aaron Grattafiori, Abhimanyu Dubey, Abhinav Jauhri, Abhinav Pandey, Abhishek Kadian, Ahmad Al-Dahle, Aiesha Letman, Akhil Mathur, Alan Schelten, Alex Vaughan, et al. The llama 3 herd of models. arXiv preprint arXiv:2407.21783, 2024.
>
> [5] Guilherme Penedo, Hynek Kydl´ıˇcek, Anton Lozhkov, Margaret Mitchell, Colin A Raffel, Leandro Von Werra, Thomas Wolf, et al. The fineweb datasets: Decanting the web for the finest text data at scale. Advances in Neural Information Processing Systems, 37:30811–30849, 2024b
>
> [6] Yang Liu, Dan Iter, Yichong Xu, Shuohang Wang, Ruochen Xu, and Chenguang Zhu. G-eval: Nlg evaluation using gpt-4 with better human alignment. arXiv preprint arXiv:2303.16634, 2023.
>
> [7] Lianmin Zheng, Wei-Lin Chiang, Ying Sheng, Siyuan Zhuang, Zhanghao Wu, Yonghao Zhuang, Zi Lin, Zhuohan Li, Dacheng Li, Eric Xing, et al. Judging llm-as-a-judge with mt-bench and chatbot arena. Advances in neural information processing systems, 36:46595–46623, 2023.

---

> ### Author Response · Authors · 2025-11-23
> **Response to Reviewer nhUc (2/n)**
>
> > (W2) The explanation of how thresholds per PC are chosen is underspecified.
>
> The threshold $t_k$ is selected according to the data budget $s_k$ on this dimension. In this work, we select the same amount of tokens from each dimension. Therefore, the threshold can be determined given the total data budget.
>
> > (W3) Interpretation of principal components remains unclear, limiting insight into what semantic attributes each dimension captures.
>
> We argue that the shift from interpretable human metrics to latent orthogonal features is a necessary trade-off for effective selection. We define the evaluation metrics (e.g., semantic complexity, knowledge depth) from a human perspective. Although these metrics seem distinct for humans, they exhibit a certain correlation to models, leading to redundancy and score bias. To address this issue, ODiS transforms these overlapping metrics into orthogonal principal components. While these components are composite features (weighted combinations of original metrics as shown in Figure 5) and thus less linguistically interpretable, they represent the true, non-redundant axes of variance in the data. By selecting from orthogonal dimensions, ODiS ensures we capture all the desired features, rather than selecting redundant samples due to overlapping human definitions.

---

> ### Author Response · Authors · 2025-11-23
> **Response to Reviewer nhUc (3/n)**
>
> > (W4) Experiments are limited to a single language and corpus (Chinese Nemotron-CC) and one model scale (1.5B), raising concerns about generality. More evaluation on larger models or multilingual datasets would strengthen conclusions.
>
> We thank the reviewer for the suggestion on generality. We address the concerns regarding language, dataset, and model scale as follows.
>
> First, we apologize for the error in the original paper that we labeled Nemotron-CC as Chinese. We would like to clarify that Nemotron-CC is a large-scale English dataset. Therefore, our experiments are conducted on the standard English pre-training dataset, not a language-specific corpus. We have corrected this description in the revision.
>
> Using a single, high-quality English corpus for data selection validation is the standard practice in recent literature due to the high computational cost of pre-training. Leading works such as DSIR [1], DoReMi [2], and MATES [3] primarily utilize single English datasets (e.g., Pile, C4, SlimPajama) for evaluation. Similarly, validating algorithms on models under 3B parameters (e.g., 1B in DoReMi [2], 1B in MATES [3], 3B in Meta-tater [4]) is the accepted norm for academic research to ensure reproducibility and computational feasibility.
>
> To further address the concern about scaling generality, we have added a new experiment training a 400M parameter model using the same setup. The results (added to Table 1 in the revision and also provided below) show that ODiS achieves significant improvements on the 400M scale, consistent with previous 1.5B results. The consistent superiority of ODiS across different model scales demonstrates the robustness of our method. While training larger models is beyond our current computational budget, the observed scaling consistency strongly suggests that ODiS effectively generalizes to larger models.
>
> | Model | Method | Arc-C | Arc-E | Hellaswag | SCIQ | PIQA | Average |
> | :--- | :--- | :---: | :---: | :---: | :---: | :---: | :---: |
> | **1.5B** | Random | 35.0 ± 1.4 | 62.7 ± 1.0 | 58.3 ± 0.5 | 85.5 ± 1.1 | 74.5 ± 1.0 | 63.2 ± 1.0 |
> | | Nemotron-HQ | 37.3 ± 1.4 | 64.6 ± 1.0 | 57.7 ± 0.5 | 83.9 ± 1.1 | 73.6 ± 1.0 | 63.4 ± 1.0 |
> | | PPL-*Top* | 37.9 ± 1.4 | 62.8 ± 1.0 | 54.7 ± 0.5 | 83.4 ± 1.2 | 74.7 ± 1.0 | 62.7 ± 1.0 |
> | | PPL-*Sample* | 36.1 ± 1.4 | 64.3 ± 1.0 | 58.4 ± 0.5 | 85.8 ± 1.1 | 74.8 ± 1.0 | 63.9 ± 1.0 |
> | | DSIR | 27.8 ± 1.3 | 48.5 ± 1.0 | 54.6 ± 0.5 | 78.5 ± 1.3 | 71.0 ± 1.1 | 56.1 ± 1.0 |
> | | PC Aver-*Top* | 35.3 ± 1.4 | 57.9 ± 1.0 | 53.3 ± 0.5 | 71.9 ± 1.4 | 74.5 ± 1.0 | 58.6 ± 1.1 |
> | | PC Aver-*Sample* | 39.2 ± 1.4 | 64.8 ± 1.0 | 58.9 ± 0.5 | 80.6 ± 1.3 | 75.1 ± 1.0 | 63.7 ± 1.0 |
> | | **ODiS** | **41.6 ± 1.4** | **66.9 ± 1.0** | **58.4 ± 0.5** | **85.6 ± 1.0** | **77.4 ± 1.0** | **66.0 ± 1.0** |
> | **400M** | Random | 26.5 ± 1.3 | 49.5 ± 1.0 | 40.6 ± 0.5 | 74.7 ± 1.4 | 67.5 ± 1.1 | 51.7 ± 1.1 |
> | | Nemotron-HQ | 29.6 ± 1.3 | 54.3 ± 1.0 | 40.7 ± 0.5 | 75.4 ± 1.4 | 67.9 ± 1.1 | 53.6 ± 1.1 |
> | | PPL-*Top* | 27.7 ± 1.3 | 50.3 ± 1.0 | 40.0 ± 0.5 | 73.0 ± 1.4 | 67.9 ± 1.1 | 51.8 ± 1.1 |
> | | PPL-*sample* | 28.2 ± 1.3 | 50.7 ± 1.0 | 41.7 ± 0.5 | 76.3 ± 1.4 | 69.9 ± 1.1 | 53.4 ± 1.1 |
> | | DSIR | 24.6 ± 1.3 | 40.7 ± 1.0 | 36.5 ± 0.5 | 66.6 ± 1.5 | 65.0 ± 1.1 | 46.7 ± 1.1 |
> | | PC-Aver-*Top* | 29.2 ± 1.3 | 48.7 ± 1.0 | 41.6 ± 0.5 | 69.9 ± 1.5 | 70.5 ± 1.1 | 52.0 ± 1.1 |
> | | PC-Aver-*Sample* | 30.1 ± 1.3 | 53.2 ± 1.0 | 43.2 ± 0.5 | 75.0 ± 1.4 | 70.1 ± 1.1 | 54.3 ± 1.1 |
> | | **ODiS** | **30.0 ± 1.3** | **54.3 ± 1.0** | **42.5 ± 0.5** | **74.5 ± 1.4** | **72.5 ± 1.0** | **54.8 ± 1.1** |
>
> [1] Sang Michael Xie, Shibani Santurkar, Tengyu Ma, and Percy S Liang. Data selection for language models via importance resampling. Advances in Neural Information Processing Systems, 36: 34201–34227, 2023b.
>
> [2] Sang Michael Xie, Hieu Pham, Xuanyi Dong, Nan Du, Hanxiao Liu, Yifeng Lu, Percy S Liang, Quoc V Le, Tengyu Ma, and Adams Wei Yu. Doremi: Optimizing data mixtures speeds up language model pretraining. Advances in Neural Information Processing Systems, 36:69798–69818, 2023a.
>
> [3] Zichun Yu, Spandan Das, and Chenyan Xiong. Mates: Model-aware data selection for efficient pretraining with data influence models. Advances in Neural Information Processing Systems, 37: 108735–108759, 2024
>
> [4] Xinlin Zhuang, Jiahui Peng, Ren Ma, Yinfan Wang, Tianyi Bai, Xingjian Wei, Qiu Jiantao, Chi Zhang, Ying Qian, and Conghui He. Meta-rater: A multi-dimensional data selection method for pre-training language models. In Proceedings of the 63rd Annual Meeting of the Association for Computational Linguistics (Volume 1: Long Papers), pp. 10856–10896, 2025

---

> ### Author Response · Authors · 2025-11-23
> **Response to Reviewer nhUc (4/n)**
>
> > (Q1) LLM Scoring Bias: Since GPT models produce the initial 11-dimensional scores, how robust is ODiS to inaccuracies or systematic biases in these annotations? Have you experimented with weak or noisy scoring sources?
>
> We argue that ODiS is robust to both bias and noise by method design. The PCA serves as a bias and noise filter. The systematic bias typically results in a correlation among the evaluation metrics, which is decoupled by PCA. Therefore, selecting the best data from each orthogonal dimension avoids bias. Moreover, PCA captures the dominant quality signal to form the top principal components with high variance, while the random noise falls into the lower-variance tail components. By selecting data from the top components, ODiS implicitly avoids GPT's random noise. Furthermore, the use of a RoBERTa scorer acts as a second layer of filtering. It learns the general trend of the GPT-labeled score while often ignoring individual outliers or random hallucinations, thereby improving the robustness of the final scores.
>
> > (Q2) Generalization: Do improvements persist for larger model sizes or multilingual corpora?
>
> Our experiments suggest that the improvements do persist and transfer across model scales. We have added experiments on a 400M model (in addition to the 1.5B model). The results show that ODiS consistently outperforms baselines at both scales. This consistent superiority across two orders of magnitude (400M to 1.5B) aligns with the consensus that data quality gains often yield larger benefits as model capacity increases. We are confident that this trend holds for larger models (e.g., 8B or larger), as high-quality, diverse data becomes even more critical for preventing saturation in large-scale training.
>
> While we acknowledge the importance of multilingual evaluation, we emphasize that our experimental setup aligns with the standard practice in the field of data selection research. Most state-of-the-art data selection algorithms validate their methods primarily on English benchmarks to isolate algorithmic efficacy from language-specific resource imbalances. Besides, ODiS is inherently language-agnostic theoretically. The 11 dimensions are universal semantic attributes, and the statistical mechanism (PCA) applies regardless of the language. Given that LLM-based scorers support multilingual input, ODiS can be directly applied to other languages without modification.

---

> ### Author Response · Authors · 2025-12-04
> **Response to Reviewer nhUc (5/n)**
>
> To further address the reviewer's concern about the generalization over larger model sizes, we have incorporated an additional experiment with a 8B model under 25B token budgets. The results are provided in Section 3.3.5 of the revised manuscript, which combines the results with 400M and 1.5B model under 100B and 200B token budgets, respectively. ODiS consistently outperforms the baselines with 8B model setting, validating its generalization to different model sizes.
>
> | Model | Method | Arc-C | Arc-E | Hellaswag | SCIQ | PIQA | Average |
> | :--- | :--- | :---: | :---: | :---: | :---: | :---: | :---: |
> | **400M** | Nemotron-HQ | 29.6 ± 1.3 | 54.3 ± 1.0 | 40.7 ± 0.5 | 75.4 ± 1.4 | 67.9 ± 1.1 | 53.6 ± 1.1 |
> | | PPL-*sample* | 28.2 ± 1.3 | 50.7 ± 1.0 | 41.7 ± 0.5 | 76.3 ± 1.4 | 69.9 ± 1.1 | 53.4 ± 1.1 |
> | | DSIR | 24.6 ± 1.3 | 40.7 ± 1.0 | 36.5 ± 0.5 | 66.6 ± 1.5 | 65.0 ± 1.1 | 46.7 ± 1.1 |
> | | PC-Aver-*Top* | 29.2 ± 1.3 | 48.7 ± 1.0 | 41.6 ± 0.5 | 69.9 ± 1.5 | 70.5 ± 1.1 | 52.0 ± 1.1 |
> | | PC-Aver-*Sample* | 30.1 ± 1.3 | 53.2 ± 1.0 | 43.2 ± 0.5 | 75.0 ± 1.4 | 70.1 ± 1.1 | 54.3 ± 1.1 |
> | | **ODiS** | **30.0 ± 1.3** | **54.3 ± 1.0** | **42.5 ± 0.5** | **74.5 ± 1.4** | **72.5 ± 1.0** | **54.8 ± 1.1** |
> | **1.5B** | Nemotron-HQ | 40.3 ± 1.4 | 67.9 ± 1.0 | 59.9 ± 0.5 | 74.5 ± 1.0 | 87.1 ± 1.1 | 65.9 ± 1.0 |
> | | PPL-Top | 37.4 ± 1.4 | 63.9 ± 1.0 | 56.4 ± 0.5 | 74.2 ± 1.0 | 84.5 ± 1.2 | 63.3 ± 1.0 |
> | | PPL-*Sample* | 38.2 ± 1.4 | 67.5 ± 1.0 | 60.5 ± 0.5 | 75.6 ± 1.0 | 88.8 ± 1.0 | 66.1 ± 1.0 |
> | | DSIR | 30.2 ± 1.3 | 50.8 ± 1.0 | 58.1 ± 0.5 | 72.2 ± 1.1 | 79.4 ± 1.3 | 58.1 ± 1.0 |
> | | PC Aver-*Top* | 36.6 ± 1.4 | 59.1 ± 1.0 | 54.8 ± 0.5 | 74.9 ± 1.0 | 72.8 ± 1.4 | 59.6 ± 1.1 |
> | | PC Aver-*Sample* | 40.0 ± 1.4 | 66.3 ± 1.0 | 61.4 ± 0.5 | 76.4 ± 0.5 | 83.4 ± 1.2 | 65.5 ± 0.9 |
> | | **ODiS** | **40.9 ± 1.4** | **68.4 ± 1.0** | **59.1 ± 0.5** | **78.4 ± 1.0** | **88.2 ± 1.0** | **67.0 ± 1.0** |
> | **8B** | Nemotron-HQ | 36.2 ± 1.4 | 62.1 ± 1.0 | 58.4 ± 0.5 | 84.3 ± 1.2 | 74.9 ± 1.0 | 63.2 ± 1.0 |
> | | PPL-sample | 36.2 ± 1.4 | 63.1 ± 1.0 | 57.9 ± 0.5 | 84.9 ± 1.1 | 73.3 ± 1.0 | 63.1 ± 1.0 |
> | | DSIR | 30.9 ± 1.4 | 49.3 ± 1.0 | 55.8 ± 0.5 | 77.8 ± 1.3 | 71.7 ± 1.1 | 57.1 ± 1.0 |
> | | PC-Aver-*Top* | 37.1 ± 1.4 | 59.1 ± 1.0 | 54.2 ± 0.5 | 72.4 ± 1.4 | 75.2 ± 1.0 | 59.6 ± 1.1 |
> | | PC-Aver-*Sample* | 41.1 ± 1.4 | 65.7 ± 1.0 | 59.5 ± 0.5 | 85.1 ± 1.1 | 75.5 ± 1.0 | 65.4 ± 1.0 |
> | | **ODiS** | **40.1 ± 1.4** | **67.2 ± 1.0** | **57.6 ± 0.5** | **85.5 ± 1.1** | **77.3 ± 1.0** | **65.5 ± 1.0** |

---

### Official Review · Reviewer_fDob · 2025-11-01

**Soundness:** 1
**Presentation:** 2
**Contribution:** 1
**Rating:** 2
**Confidence:** 4

**Summary:**

The authors propose a new data (pretraining) selection algorithm that (1) chooses a space of scores (in this case, scores by GPT, although the model is not mentioned), (2) performs PCA to identify independent directions, (3) trains a RoBERTa classifier to regress the PCA-transformed scores, (4) select a dataset based on the transformed scores, a joint score threshold and a total data budget.

**Strengths:**

- The paper is well written
- The introduction is clear

**Weaknesses:**

1. I think a fundamental limitation of data selection research is that the scale with which it is conducted is too small. The difference between the best 100B tokens and the average 100B tokens is massive, and thus data selection methods can make a huge difference. But once one is forced to go to 50T-100T tokens, there simply aren’t 50T high quality tokens and 50T low quality tokens to separate. This is the point made by https://openaccess.thecvf.com/content/CVPR2024/html/Goyal_Scaling_Laws_for_Data_Filtering--_Data_Curation_cannot_be_Compute_CVPR_2024_paper.html. Consequently, I am skeptical that this method will make a difference at larger token budgets.

2. Section 2 is supposed to serve as motivation for the proposed algorithm ODiS, but the section is poorly explained with many concerns, and the pointers to learn more are incorrect and duplicated ("Sections 2.3, 3.1, and 3.1"). Without the motivation, the paper loses wind in its sails. As best as I can tell, Figure 1 doesn't show what the text claims it shows, Figure 2a looks like low quality noise and Figure 2b lacks context to be understood.

3. Section 3's demonstration of ODiS's superiority comes across as weak evidence to me. There are no confidence intervals or error bars (over samples in the benchmarks) to identify whether any of these scores are meaningfully different. Additionally, ODiS has so many hyperparameters (e.g., $\tau$, the joint score threshold $t$, how annotations are obtained, how the RoBERTA-based scorers are trained) that there's simply no way for the reader to know whether these scores are genuine improvements or p-hacked.

**Questions:**

## Title

- The title is a bit generic. It doesn’t communicate ODiS as a name or the key idea of ODiS

## Introduction

- Straightforwardly written - nice!
- nit: Use \citet{} and \citep{} as appropriate. For example, line 54/55 should be \citep{zhuang2025…, liu2025…,bai2025…}. This avoids the double parentheses

## Section 2.1

- line 102: nit: “Sections 2.3, 3.1, and 3.1” repeat 3.1 twice
- Section 2.1: I am confused by the methodology here and what exact claim is being made. The text says to look at 2.3, 3.1 and 3.1 for details, but as best as I can tell, those sections do not provide details on Figure 1
- Section 2.1: The key claim in this section “From Figure 1, we can observe that the data with the highest score performs the worst, while sampling data from a broader range leads to improvement” does not seem to be substantiated in Figure 1 at all. As best as I can tell, Figure 1 doesn’t show scores. Instead, it shows that a larger dataset to search through likely yields better performance across 4/5 evals while the fifth (PIQA) is noise.
- Figure 1: Where are the confidence intervals / standard errors? Each score is averaged over many samples - plot the uncertainty so we can know which (if any) are meaningfully different.
- Figure 1: Add a more descriptive caption. Make it easy for a lazy reader. I shouldn’t have to hunt through the paper to understand what was done, how to interpret these results, what your point is.
- Figure 2a: I have no idea what this figure is meant to communicate. It looks like noise.
- Figure 2b: I again do not know how to interpret this figure. What does “Average-100B” mean in this case? Are we averaging over different subsets of 100B tokens? If so, how is that averaging done? Are you creating multiple histograms and then averaging each bucket height?
- Line 140: Where is the citation for m3e?
- Line 121-122: “ whereas top-scored data is relatively homogeneous, which explains the performance degradation of the top-scored data.” Where is this shown? I am confused by where scores are being visualized?

## Section 2.3

- The criticism about directly optimizing for downstream tasks is valid, but I’m not sure I understand how creating 11 proxy dimensions is anything more than a less direct proxy.

## Section 3.1

- line 257: “It is a large-scale Chinese dataset” To the best of my knowledge, Nemotron CC is not a Chinese dataset?

## Section 3.2

- Table 1: Where are the confidence intervals over samples in the benchmarks? How can we say whether any of these scores are meaningfully different?

---

> ### Author Response · Authors · 2025-11-23
> **Response to Reviewer fDob (1/n)**
>
> Thank you for your valuable feedback to help us improve our paper. We have revised our paper based on your feedback. We detail our response below and please kindly let us know if our response addresses your concerns.
>
> > (W1) I think a fundamental limitation of data selection research is that the scale with which it is conducted is too small. The difference between the best 100B tokens and the average 100B tokens is massive, and thus data selection methods can make a huge difference. But once one is forced to go to 50T-100T tokens, there simply aren’t 50T high quality tokens and 50T low quality tokens to separate. This is the point made by https://openaccess.thecvf.com/content/CVPR2024/html/Goyal_Scaling_Laws_for_Data_Filtering--_Data_Curation_cannot_be_Compute_CVPR_2024_paper.html. Consequently, I am skeptical that this method will make a difference at larger token budgets.
>
> We thank the reviewer for raising the discussion regarding the effectiveness of data selection at a large scale. We would like to argue that our method remains highly effective and provide the following clarification to address this concern.
>
> First, we think that the paper you provided doesn’t conflict with the point that the data selection method cannot make a big difference at the 50T-100T scale. The cited paper primarily examines the trade-off between data quality and data repetition. Their findings demonstrate that training a model with a repeated small subset of high-quality data (e.g., seeing 10B tokens 10 times) yields diminishing returns compared to a larger but unique dataset (e.g., seeing 100B tokens once). This does not negate the value of data selection. Instead, it highlights the need for selecting high-quality data without excessive repetition.
>
> In addition, we believe that even at the scale of 50T tokens (although we as LLM practitioners all know that among the existing LLM with revealed corpus information, the largest training corpus contains nearly 30T tokens), data selection remains necessary for the need of noise filtering and data synthesis in the data curation pipeline of modern large language models like Llama, Qwen, DeepSeek, and so on. For noise filtering, raw web data (e.g., CommonCrawl) contains a lot of noise and repeated content. We should always select from a larger raw pool and remove the toxic and incoherent text, which is done during the training of primary LLMs and the construction of large-scale datasets (e.g., LLAMA 3 utilizes 15T tokens with filtering pipelines [1] and FineWeb dataset is constructed with filtering from PB level documents to 15T tokens [2]). For data synthesis, as pointed out in recent works, when raw high-quality data is exhausted, data selection methods are crucial for identifying high-quality reference data to generate synthetic ones, thereby solving the scarcity issue mentioned by the reviewer.
>
> Finally, our experimental setting (100B tokens) aligns with established benchmarks in recent top-tier literature, such as Meta-Rater (30B budget, [3]) and MATES (25B tokens, [4]), and Multi-actor collaboration data selection (30B tokens, [5]). We believe that this scale provides robust validation of the effectiveness of our proposed method, which can then be extended to larger scales.
>
> [1] Aaron Grattafiori, Abhimanyu Dubey, Abhinav Jauhri, Abhinav Pandey, Abhishek Kadian, Ahmad Al-Dahle, Aiesha Letman, Akhil Mathur, Alan Schelten, Alex Vaughan, et al. The llama 3 herd of models. arXiv preprint arXiv:2407.21783, 2024.
>
> [2] Guilherme Penedo, Hynek Kydl´ıˇcek, Loubna Ben allal, Anton Lozhkov, Margaret Mitchell, Colin Raffel, Leandro Von Werra, and Thomas Wolf. The fineweb datasets: Decanting the web for the finest text data at scale. In The Thirty-eight Conference on Neural Information Processing Systems Datasets and Benchmarks Track, 2024a.
>
> [3] Xinlin Zhuang, Jiahui Peng, Ren Ma, Yinfan Wang, Tianyi Bai, Xingjian Wei, Qiu Jiantao, Chi Zhang, Ying Qian, and Conghui He. Meta-rater: A multi-dimensional data selection method for pre-training language models. In Proceedings of the 63rd Annual Meeting of the Association for Computational Linguistics (Volume 1: Long Papers), pp. 10856–10896, 2025
>
> [4] Zichun Yu, Spandan Das, and Chenyan Xiong. Mates: Model-aware data selection for efficient pretraining with data influence models. Advances in Neural Information Processing Systems, 37: 108735–108759, 2024
>
> [5] Tianyi Bai, Ling Yang, Zhen Hao Wong, Fupeng Sun, Xinlin Zhuang, Jiahui Peng, Chi Zhang, Lijun Wu, Qiu Jiantao, Wentao Zhang, et al. Efficient pretraining data selection for language models via multi-actor collaboration. In Proceedings of the 63rd Annual Meeting of the Association for Computational Linguistics (Volume 1: Long Papers), pp. 9465–9491, 2025.

---

> ### Author Response · Authors · 2025-11-23
> **Response to Reviewer fDob(2/n)**
>
> > (W2) Section 2 is supposed to serve as motivation for the proposed algorithm ODiS, but the section is poorly explained with many concerns, and the pointers to learn more are incorrect and duplicated ("Sections 2.3, 3.1, and 3.1"). Without the motivation, the paper loses wind in its sails. As best as I can tell, Figure 1 doesn't show what the text claims it shows, Figure 2a looks like low quality noise and Figure 2b lacks context to be understood.
>
> We thank the reviewer for pointing out the clarity issues in Section 2. We acknowledge that the motivation could be better articulated and more explanation is required in this part. We have revised Section 2 by incorporating more explanations regarding the experiment and the interpretation of results to improve readability in the revised version. Moreover, we offer the following clarifications to address your concern.
>
> The core motivation for this paper is to point out the bias of the score-based data selection method and propose to mitigate the evaluation bias through orthogonality. Figure 1 is designed to demonstrate the selection bias. The experiment is conducted with a 1.5B model under a 100B token budget with varying data selection ranges. We would like to clarify that the x-axis demonstrates the sizes of the top-ranked data candidate pool, not the training data size. This setup allows us to investigate the model performance in relation to the data quality (higher scores in smaller pools). The results show that directly training the model with the highest score yields a suboptimal performance, while a larger candidate pool with lower scores achieves better results. This phenomenon indicates that there is a bias in the score-based data selection method. Then, Figure 2a and 2b further analyze where the bias comes from.
>
> Figure 2a is designed to investigate data diversity of different candidate pools through the embedding distribution visualization in a compressed space with UMAP. We understand that the scatter plot might appear unstructured at first glance. However, the points that appear as noise (i.e., points from Pool size-500B) represent a broader semantic coverage, while the top-scored data points (i.e., points from Pool size-100B) cluster tightly in specific regions. This demonstrates that score-based methods lead to feature collapse (loss of diversity), leaving vast areas of the semantic space uncovered.
>
> Figure 2b is also designed to illustrate data diversity. We acknowledge that the explanations for the labels are missing. To make the figure easier to understand, we have revised the labels to “Pool size NB”, which directly indicates the distinction between different data sources.
>
> Based on the analysis in Section 2.1, we would like to conclude that the score-based methods only focus on the quality while overlooking diversity. Therefore, we try to preserve data diversity during high-quality data selection by creating several orthogonal dimensions and selecting the best data from each of them.
>
> > (W3) Section 3's demonstration of ODiS's superiority comes across as weak evidence to me. There are no confidence intervals or error bars (over samples in the benchmarks) to identify whether any of these scores are meaningfully different.
>
> We appreciate the reviewer’s advice regarding statistical significance. In the revised version, we have reported the standard errors for all benchmark results. Specifically, the standard errors across the benchmarks range from 0.005 to 0.015. This variation is relatively small compared to the performance margin of ODiS over the baselines (which typically exceeds 0.03). Although we observe that while some baselines may achieve comparable results on specific tasks, ODiS consistently outperforms them on the average performance. Given that the performance gap on the averaged score is consistently larger than the standard error (by a factor of 2), we confirm that the superiority of ODiS is statistically significant and not due to random variance.

---

> ### Author Response · Authors · 2025-11-23
> **Response to Reviewer fDob (3/n)**
>
> > (W4) Additionally, ODiS has so many hyperparameters (e.g., , the joint score threshold , how annotations are obtained, how the RoBERTA-based scorers are trained) that there's simply no way for the reader to know whether these scores are genuine improvements or p-hacked.
>
> We appreciate the reviewer’s concern regarding hyperparameter sensitivity. We respectfully clarify that ODiS does not rely on extensive hyperparameter tuning, and the results are not outcomes of p-hacking, with the explanation as follows:
>
> The selection threshold is not a tunable hyperparameter. Instead, it is dynamically determined by the data budget. For a budget of $N$ tokens and $D$ dimensions, ODiS simply selects the top-$\frac{N}{D}$ tokens from each dimension. This is a deterministic rule, not a tuned value.
>
> The training of RoBERTa-based scorers follows the standard regression practice. Default hyperparameters (e.g., batch sizes and epochs) along with the adjustments of the learning rates are enough to carry out the training. We also make sure that the data sequence of experiments are the same. This pipeline is widely adopted in existing works like FineWeb [1] and QuRating [2], ensuring that our setup aligns with community norms rather than custom engineering. To further ensure transparency, we have added the details of Roberta training in Appendix D.
>
> The GPT annotations are derived from a fixed set of prompts (provided in Appendix B) based on the definitions of the 11 dimensions. These are conceptual definitions, not numerical parameters to be tuned.
>
> Given that our results were obtained using the above deterministic pipelines and standard settings without extensive grid search, and that the standard error of the results are relatively smaller, we are confident that the performance improvement between the proposed algorithm and the baselines is genuine and robust.
>
> [1] Guilherme Penedo, Hynek Kydlivek, Loubna Ben allal, Anton Lozhkov, Margaret Mitchell, Colin Raffel, Leandro Von Werra, and Thomas Wolf. The fineweb datasets: Decanting the web for the finest text data at scale. In The Thirty-eight Conference on Neural Information Processing Systems Datasets and Benchmarks Track, 2024a.
>
> [2] Alexander Wettig, Aatmik Gupta, Saumya Malik, and Danqi Chen. Qurating: Selecting high-quality data for training language models. arXiv preprint arXiv:2402.09739, 2024.
>
> > (Q1) The title is a bit generic. It doesn’t communicate ODiS as a name or the key idea of OdiS
>
> We thank the reviewer for the feedback on the title. Our intention for the current title was to emphasize the generalizable paradigm shift proposed in this work: moving from a weighted-sum evaluation to a decoupled, multi-dimensional perspective on data selection. We believe the current phrasing captures this high-level contribution. We view ODiS as the concrete realization of this broader framework, and we initially prioritized highlighting the conceptual contribution over the specific algorithm name.
>
> > (Q2) Introduction: Use \citet{} and \citep{} as appropriate. For example, line 54/55 should be \citep{zhuang2025…, liu2025…,bai2025…}. This avoids the double parentheses
>
> Thanks for your advice. We have updated the citation in the revised version.
>
> > (Q3.1) line 102: nit: “Sections 2.3, 3.1, and 3.1” repeat 3.1 twice
>
> We thank the reviewer for this advice. We agree that the repetition was confusing. We have rephrased the sentence in the revised version to refer to the contents of Section 3.1 more clearly and removed the redundancy.
>
> > (Q3.2) Section 2.1: I am confused by the methodology here and what exact claim is being made. The text says to look at 2.3, 3.1 and 3.1 for details, but as best as I can tell, those sections do not provide details on Figure 1
>
> We thank the reviewer for raising this concern. Section 2.1 is meant to examine the cause of the bias of score-based data selection method through a case study and serves as the motivation for the following methodology. Figure 1 is designed to demonstrate that models trained with top-scored data yield suboptimal performance while sampling from a broader scope is necessary to recover performance. Therefore, after scoring data, we trained a 1.5B parameter model using a fixed budget of 100B tokens with varying candidate pool sizes. The x-axis is the size of the candidate pool (ranging from top-100B to top-900B scored data). The results demonstrate that directly selecting the highest-scored data (e.g., the leftmost point) yields suboptimal performance. Expanding the candidate pool (moving right) recovers performance, indicating that traditional score-based selection introduces bias by sacrificing data diversity, which motivates our ODiS approach.
>
> Since the training details and the benchmark selection are not the core here and are similar to the main experiments in the following section, we describe them briefly and refer the readers to the corresponding sections for the details.

---

> ### Author Response · Authors · 2025-11-23
> **Response to Reviewer fDob (4/n)**
>
> > (Q3.3) Section 2.1: The key claim in this section “From Figure 1, we can observe that the data with the highest score performs the worst, while sampling data from a broader range leads to improvement” does not seem to be substantiated in Figure 1 at all. As best as I can tell, Figure 1 doesn’t show scores. Instead, it shows that a larger dataset to search through likely yields better performance across 4/5 evals while the fifth (PIQA) is noise.
>
> We appreciate the reviewer for highlighting this confusion. We would like to clarify that Figure 1 implicitly shows the relationship between scores and performance through the construction of the candidate pools. The candidate pools (shown in x-axis) are created by labeling the corpus with scores and selecting the top-K tokens (e.g., top-100B). Therefore, there is a direct relationship between the candidate pool size and the data quality score: the candidate pool with 100B tokens (leftmost) consists of the highest-scored tokens, while the candidate pool with 900B tokens (rightmost) includes tokens with lower scores, indicating a lower average score compared to the 100B pool. Since we sample a fixed budget (e.g., 100B tokens) for all experiments, the improvement observed as we move from 100B to 900B (left to right) demonstrates that training on data with highest scores (the top-100B) performs worse than training on data with lower average scores but higher diversity (the top-900B). Moreover, the benchmark selection is described in Section 3.1. We select the benchmarks through the experiments over different token budgets to see whether the benchmark is sensitive to the model training. The PIQA appears noisy here mainly because the models do not exhibit significant distinctions under this benchmark.
>
>
> > (Q3.4) Figure 1: Where are the confidence intervals / standard errors? Each score is averaged over many samples - plot the uncertainty so we can know which (if any) are meaningfully different.
>
> Thanks for your advice. We have added the standard errors to the figure. The performance difference is relatively large compared to the standard errors under most of the benchmarks. Therefore, we believe that these results are valid.
>
> > (Q3.5) Figure 1: Add a more descriptive caption. Make it easy for a lazy reader. I shouldn’t have to hunt through the paper to understand what was done, how to interpret these results, what your point is.
>
> Thanks for your advice. We have incorporated more details (e.g., training details and interpretation) into the caption of Figure 1 to make it more self-contained. We hope that our explanation and the revised caption could address your concerns regarding Figure 1.
>
> > (Q3.6) Figure 2a: I have no idea what this figure is meant to communicate. It looks like noise.
>
> We thank the reviewer for raising the clarity issue. Figure 2a is designed to illustrate the embedding diversity of different candidate pools, by visualizing the semantic space via UMAP of document embeddings.  The figure distinguishes between data selected from narrow scopes (Top-100B) versus broader scopes (Top-500B). It shows that the Top-100B data is limited to specific regions (indicating feature collapse or limited diversity), whereas the broader selection (Top-500B) covers the space more widely. Although the scatter plot might appear unstructured at first glance, it represents high semantic coverage. Ideally, a diverse dataset should uniformly cover a larger embedding space. Therefore, it validates our claim that expanding the selection scope recovers the semantic diversity lost by score-based filtering.
>
> > (Q3.7) Figure 2b: I again do not know how to interpret this figure. What does “Average-100B” mean in this case? Are we averaging over different subsets of 100B tokens? If so, how is that averaging done? Are you creating multiple histograms and then averaging each bucket height?
>
> We apologize for the ambiguity in the label. The “Average” is not averaging histograms or bucket heights, but it refers to selecting data based on the average score from multiple evaluation metrics (as described in Section 2.1). Moreover, the token amount represents the size of the candidate pool constructed from top-score tokens. To make the label more understandable, we have revised the labels to “Pool size - NB” where N is the number of tokens.
>
> > (Q3.8) Line 140: Where is the citation for m3e?
>
> Thanks for pointing out the citation missing. M3e is an embedding model with the following citation. We have added the citation to the revised version.
>
> [1] He sicheng Wang Yuxin, Sun Qingxuan. M3e: Moka massive mixed embedding model,

---

> ### Author Response · Authors · 2025-11-23
> **Response to Reviewer fDob (5/n)**
>
> > (Q3.9) Line 121-122: “ whereas top-scored data is relatively homogeneous, which explains the performance degradation of the top-scored data.” Where is this shown? I am confused by where scores are being visualized?
>
> We would like to make the clarification about the claim. The scores are implicitly represented by the selection scope (e.g., 100B vs. 900B), since we select the data through the top-k method and the training budget is fixed. The claim can be obtained through connecting evidence from Figure 1 and Figure 2. Figure 1 demonstrates that the top-scored data (e.g., the top-100B tokens) will lead to performance degradation, and Figure 2 illustrates that these data contain less diversity (e.g., homogeneous) compared with data from a larger score scope (e.g., the top-500B tokens). Since the top-scored data has the highest quality (which is quantified by the scores), the diversity lost explains the performance degradation.
>
> > (Q4) Section 2.3: The criticism about directly optimizing for downstream tasks is valid, but I’m not sure I understand how creating 11 proxy dimensions is anything more than a less direct proxy.
>
> We thank the reviewer for raising the concern about the proxy in data evaluation. The fundamental difference between the evaluation dimension and the “less direct proxy” lies in the general capability of the LLMs. We would like LLMs to exhibit certain capabilities, such as fluent text generation, and reasoning ability. These general capabilities are based on human perception, which are difficult to characterize and model. Directly optimizing a task or proxy is like learning a specific distribution, while evaluation through dimensions is optimized towards the ultimate capability.
>
> > (Q5) Section 3.1: line 257: “It is a large-scale Chinese dataset” To the best of my knowledge, Nemotron CC is not a Chinese dataset?
>
> Thanks for pointing out the mistake. The Nemotron-CC is indeed an English dataset. We have corrected this in the revised version.

---

> ### Author Response · Authors · 2025-11-23
> **Response to Reviewer fDob (6/n)**
>
> > (Q6) Section 3.2: Table 1: Where are the confidence intervals over samples in the benchmarks? How can we say whether any of these scores are meaningfully different?
>
> We thank the reviewer for the issue of statistical significance. We have incorporated the standard error of benchmarks in all the experiment results, and the Table 1 in the revised manuscript is provided as follows for your convenience. Since the standard error is much smaller than the performance gap, we believe that our results can effectively validate the performance of different data selection algorithms.
>
>
> | Model | Method | Arc-C | Arc-E | Hellaswag | SCIQ | PIQA | Average |
> | :--- | :--- | :---: | :---: | :---: | :---: | :---: | :---: |
> | **1.5B** | Random | 35.0 ± 1.4 | 62.7 ± 1.0 | 58.3 ± 0.5 | 85.5 ± 1.1 | 74.5 ± 1.0 | 63.2 ± 1.0 |
> | | Nemotron-HQ | 37.3 ± 1.4 | 64.6 ± 1.0 | 57.7 ± 0.5 | 83.9 ± 1.1 | 73.6 ± 1.0 | 63.4 ± 1.0 |
> | | PPL-*Top* | 37.9 ± 1.4 | 62.8 ± 1.0 | 54.7 ± 0.5 | 83.4 ± 1.2 | 74.7 ± 1.0 | 62.7 ± 1.0 |
> | | PPL-*Sample* | 36.1 ± 1.4 | 64.3 ± 1.0 | 58.4 ± 0.5 | 85.8 ± 1.1 | 74.8 ± 1.0 | 63.9 ± 1.0 |
> | | DSIR | 27.8 ± 1.3 | 48.5 ± 1.0 | 54.6 ± 0.5 | 78.5 ± 1.3 | 71.0 ± 1.1 | 56.1 ± 1.0 |
> | | PC Aver-*Top* | 35.3 ± 1.4 | 57.9 ± 1.0 | 53.3 ± 0.5 | 71.9 ± 1.4 | 74.5 ± 1.0 | 58.6 ± 1.1 |
> | | PC Aver-*Sample* | 39.2 ± 1.4 | 64.8 ± 1.0 | 58.9 ± 0.5 | 80.6 ± 1.3 | 75.1 ± 1.0 | 63.7 ± 1.0 |
> | | **ODiS** | **41.6 ± 1.4** | **66.9 ± 1.0** | **58.4 ± 0.5** | **85.6 ± 1.0** | **77.4 ± 1.0** | **66.0 ± 1.0** |
> | **400M** | Random | 26.5 ± 1.3 | 49.5 ± 1.0 | 40.6 ± 0.5 | 74.7 ± 1.4 | 67.5 ± 1.1 | 51.7 ± 1.1 |
> | | Nemotron-HQ | 29.6 ± 1.3 | 54.3 ± 1.0 | 40.7 ± 0.5 | 75.4 ± 1.4 | 67.9 ± 1.1 | 53.6 ± 1.1 |
> | | PPL-*Top* | 27.7 ± 1.3 | 50.3 ± 1.0 | 40.0 ± 0.5 | 73.0 ± 1.4 | 67.9 ± 1.1 | 51.8 ± 1.1 |
> | | PPL-*sample* | 28.2 ± 1.3 | 50.7 ± 1.0 | 41.7 ± 0.5 | 76.3 ± 1.4 | 69.9 ± 1.1 | 53.4 ± 1.1 |
> | | DSIR | 24.6 ± 1.3 | 40.7 ± 1.0 | 36.5 ± 0.5 | 66.6 ± 1.5 | 65.0 ± 1.1 | 46.7 ± 1.1 |
> | | PC-Aver-*Top* | 29.2 ± 1.3 | 48.7 ± 1.0 | 41.6 ± 0.5 | 69.9 ± 1.5 | 70.5 ± 1.1 | 52.0 ± 1.1 |
> | | PC-Aver-*Sample* | 30.1 ± 1.3 | 53.2 ± 1.0 | 43.2 ± 0.5 | 75.0 ± 1.4 | 70.1 ± 1.1 | 54.3 ± 1.1 |
> | | **ODiS** | **30.0 ± 1.3** | **54.3 ± 1.0** | **42.5 ± 0.5** | **74.5 ± 1.4** | **72.5 ± 1.0** | **54.8 ± 1.1** |

---

> ### Comment · Reviewer_fDob · 2025-11-28
> **Let's Make Sure I'm Tracking Because Right Now I'm Still Confused**
>
> I appreciate the authors improving Figure 1 and the corresponding text. I want to make sure that I understand it before we proceed because I am still finding both hard to understand.
>
> > Specifically, we select the data with the top-k scores at different scales, ranging from 100B to 900B tokens.
>
> This sentence is difficult for me to follow. What I understand you are saying is: We consider sets of data ranging from 100B tokens to 900B tokens (your "candidate pools"). For each set size, we rank the data following Wettig et al (2024) and then take the highest scoring 100B data subset. Is this correct?
>
> > while sampling data from a broader range (e.g., the candidate pool size larger than 100B) leads to improvement
>
> In the cases that the set of candidate data is larger than 100B tokens, how do you subsample? I cannot find this explained in the main text or the Figure 1 caption. You have your ordering of all the, say, 900B tokens - then what?
>
> In the Figure 1 caption:
>
> > The x-axis denotes the Top-K candidate pool size ranked by average scores.
>
> I am confused by this. My understanding is that the _candidate_ dataset size (e.g., 200B, 300B, ..., 900B) does not depend on the ranking of average scores. Is this correct? If so, why does the caption say that the candidate dataset size is ranked by average scores? Ranking is done subsequently and needn't be mentioned for the x-axis.
>
> > The leftmost point (100B) represents training on the highest-scored data, while moving right implies including lower-scored data.
>
> The "including lower-scored data" is ambiguously phrased. I think you mean that the lower scored data is included in the pool of candidate data? If so, say that simply: "Moving right implies a larger pool of candidate data to score and subsample". If I misunderstood, please clarify.
>
> Let's make sure I'm following please, so I can better provide feedback on the paper.

---

> > ### Author Response · Authors · 2025-11-28
> > **Response to Reviewer fDob (7/n)**
> >
> > Thank you for your follow-up and detailed feedback. We realize that our previous explanation regarding “training set” and the “candidate date pool” in Figure 1’s setting was not sufficiently clear. Below, we first provide the details of the data selection procedure of Figure 1’s setting and then provide a point-to-point reply to your concerns.
> >
> > The training data in Figure 1 as well as all the experiments in the paper are selected with the following steps: scoring, sorting, candidate pool selection, uniform sampling within candidate pool. Specifically for Figure 1’s setting, we first score every document within Nemotroc-CC corpus (several TB tokens) with the multi-dimensional metrics and then average them to obtain an averaged score (as described in line 97). Then, the data is sorted from high to low according to the average score. During the candidate pool selection, we adopted different top-k thresholds to filter out Top-N billion tokens from the sorted corpus to construct candidate pool with desired sizes (as described in line 100-101). For example, using a 0-5 score system where higher score indicates better quality, the 100B candidate pool consists of data with scores exceeding 3.18, while the 900B candidate pool consists of data with scores exceeding 2.9. Finally, we uniformly sample 100B tokens from the candidate pool to train a model under a 100B token budget (as described in the caption of Figure 1).
> >
> > > (Q1) “Specifically, we select the data with the top-k scores at different scales, ranging from 100B to 900B tokens.” This sentence is difficult for me to follow. What I understand you are saying is: We consider sets of data ranging from 100B tokens to 900B tokens (your "candidate pools"). For each set size, we rank the data following Wettig et al (2024) and then take the highest scoring 100B data subset. Is this correct?
> >
> > The interpretation is partially correct regarding the ranking, but incorrect regarding the final data selection. We rank the data with average scores following Wettig et al (2024). However, we do not select the highest scoring 100B data subset again from each candidate pool, as this would result in the same 100B subset for each setting. Instead, we uniformly sample 100B tokens from each candidate pool for the training. For example, if candidate pool size is 100B, we will take all the tokens. If the candidate pool size is 900B, we will uniformly select 1 out of 9 tokens.
> >
> > > (Q2) “while sampling data from a broader range (e.g., the candidate pool size larger than 100B) leads to improvement” In the cases that the set of candidate data is larger than 100B tokens, how do you subsample? I cannot find this explained in the main text or the Figure 1 caption. You have your ordering of all the, say, 900B tokens - then what?
> >
> > We randomly subsample the candidate pool data to 100B. In the context of pre-training with a fixed data budget, this is implemented as follows: the model will stream the data randomly from the candidate pool for the duration of the data budget (i.e., for a 100B data pool the training lasts for 1 epoch, while for a 900B data pool, the training only utilizes 1 of 9 data randomly). We have incorporated the wording “randomly” to the caption of Figure 1.
> >
> > > (Q3) “In the Figure 1 caption: The x-axis denotes the Top-K candidate pool size ranked by average scores.” I am confused by this. My understanding is that the candidate dataset size (e.g., 200B, 300B, ..., 900B) does not depend on the ranking of average scores. Is this correct? If so, why does the caption say that the candidate dataset size is ranked by average scores? Ranking is done subsequently and needn't be mentioned for the x-axis.
> >
> > The candidate pools are selected with the top-k method with different thresholds from Nemotron-CC dataset (as described in line 98-99). And the ranking is done before the candidate pool construction (as described in line 100-101). Specifically, the Nemotron-CC dataset contains data with several TB tokens, and we select the subset according to the averaged scores (i.e., a 3.18 threshold to obtain the 100B candidate pool, and a 2.8 threshold to obtain the 900B candidate pool). Therefore, there is an implicit relationship between candidate pool size and the averaged data score in the candidate pool (e.g., the larger the pool is, the lower the average score is).

---

> > ### Author Response · Authors · 2025-11-28
> > **Response to Reviewer (8/n)**
> >
> > > (Q4) “The leftmost point (100B) represents training on the highest-scored data, while moving right implies including lower-scored data.” The "including lower-scored data" is ambiguously phrased. I think you mean that the lower scored data is included in the pool of candidate data? If so, say that simply: "Moving right implies a larger pool of candidate data to score and subsample". If I misunderstood, please clarify.
> >
> > We would like to clarify the construction of the candidate data pool. In our setting, scoring is completed before the candidate data pool construction (i.e., the candidate data pool is constructed based on sorted scored data). Moreover, the candidate pool size and average data score in the pool are linked. Since we construct the pool with top-k methods and different thresholds, a larger pool will include data with lower scores. Therefore, expanding the pool size will result in a decrease of the average data score in the pool.
> >
> > We would like to further clarify that the core purpose of Figure 1 is to demonstrate that the subset with the highest score data might not yield the best performance. Counter-intuitively, selecting data from a larger scope and then uniformly sampling from it (i.e., first select a candidate pool larger than 100B and then uniformly sample from it) will have a better performance. The reminder of Section 2.1 provides analysis to this phenomenon.
> >
> > We hope the above explanations clarify the setup of Figure 1. Please let us know if any ambiguity remains.

---

> ### Author Response · Authors · 2025-12-04
> **Response to Reviewer fDob (9/n)**
>
> To further address the reviewer's concern about the effectiveness of the proposed ODiS under larger data budgets, we have added an additional experiment with 1.5B model under 200B token budget. The results are summarized in Section 3.3.5 of the revised manuscript. ODiS consistently outperforms baselines, validating the importance of data selection under a larger data budget.
>
> | Model | Method | Arc-C | Arc-E | Hellaswag | SCIQ | PIQA | Average |
> | :--- | :--- | :---: | :---: | :---: | :---: | :---: | :---: |
> | **1.5B** | Nemotron-HQ | 40.3 ± 1.4 | 67.9 ± 1.0 | 59.9 ± 0.5 | 74.5 ± 1.0 | 87.1 ± 1.1 | 65.9 ± 1.0 |
> | | PPL-Top | 37.4 ± 1.4 | 63.9 ± 1.0 | 56.4 ± 0.5 | 74.2 ± 1.0 | 84.5 ± 1.2 | 63.3 ± 1.0 |
> | | PPL-*Sample* | 38.2 ± 1.4 | 67.5 ± 1.0 | 60.5 ± 0.5 | 75.6 ± 1.0 | 88.8 ± 1.0 | 66.1 ± 1.0 |
> | | DSIR | 30.2 ± 1.3 | 50.8 ± 1.0 | 58.1 ± 0.5 | 72.2 ± 1.1 | 79.4 ± 1.3 | 58.1 ± 1.0 |
> | | PC Aver-*Top* | 36.6 ± 1.4 | 59.1 ± 1.0 | 54.8 ± 0.5 | 74.9 ± 1.0 | 72.8 ± 1.4 | 59.6 ± 1.1 |
> | | PC Aver-*Sample* | 40.0 ± 1.4 | 66.3 ± 1.0 | 61.4 ± 0.5 | 76.4 ± 0.5 | 83.4 ± 1.2 | 65.5 ± 0.9 |
> | | **ODiS** | **40.9 ± 1.4** | **68.4 ± 1.0** | **59.1 ± 0.5** | **78.4 ± 1.0** | **88.2 ± 1.0** | **67.0 ± 1.0** |

---

### Author Response · Authors · 2025-12-04
**Summary of the rebuttal for Area Chair**

Dear Area Chair,

As the discussion stage concludes, we would like to summarize the major revisions and our engagement with the reviewers for your convenience. We have revised the manuscript to address all the concerns from the reviewers.

- Additional scalability experiments (Section 3.3.5). To validate the scalability of ODiS on different model and data budget settings, we have incorporated experiments on 400M and 8B models. ODiS consistently outperforms baselines in both settings.

- Expanded baselines (Section 3.2). We have added three baselines that adopt a similar separate-then-select paradigm as our method, and the results further highlight the performance of ODiS and the necessity of orthogonal dimension construction.

- Cost and robustness analysis (Appendix E). We have provided a comprehensive analysis of computational cost compared to the baselines, which demonstrates that the proposed method is efficient while achieving the best performance. We also addressed concerns regarding potential bias in GPT-based scoring.

- Clarification

  - Case study (Section 2.1). In response to Reviewer fDob, we have refined the captions and descriptions to improve the clarity of the case study.

  - Methodology. We provided detailed explanations of the ODiS algorithm to the Reviewers, including why we choose PCA for orthogonalization, interpretation of orthogonal dimensions, and parameter selection.

  - Interpretation of the experiment results. We have added the standard error to the results to indicate their effectiveness.

  - Correction of typos. We have revised the manuscript to correct the typos, especially the description of the Nemotron-CC dataset.

We believe that the concerns from Reviewer KEDT and wRju are well addressed during the discussion through their responses where the Reviewer KEDT has raised the score to 6 and Reviewer wRju indicated that the scored will be raised if the results of additional baselines are provided. As for Reviewer fDob, we have provided point-to-point responses to the latest inquiries. We remain confident that the revised manuscript will address all the issues regarding understanding. Although Reviewer nhUc hasn’t responded to our rebuttal yet, we have thoroughly addressed the initial comments in the revised manuscript.

We sincerely appreciate the time and effort dedicated by you and the reviewers to improve our work. We believe that the extensive additional experiments and clarifications have enhanced the effectiveness of ODiS and significantly strengthened the paper’s contribution. We hope that the revised manuscript meets the high standards of ICLR, and we look forward to your positive decision.

Best regards,

The Authors

---

### Meta-Review · Area_Chair_ZYUA · 2026-01-02

**Summary:**

This paper proposes ODiS, a diversity-aware data selection framework for LLM pretraining that decorrelates multiple quality metrics via PCA and selects data from orthogonal dimensions, with the goal of jointly preserving quality and diversity.

While the idea of addressing metric correlation in data selection is interesting, the primary concern is that the empirical evidence does not convincingly establish that orthogonalization itself is the key driver of the reported gains, as opposed to simply leveraging multiple diverse signals or broader sampling strategies. Despite substantial rebuttal efforts and added experiments, the methodology remains complex and difficult to interpret, with limited clarity on the semantic meaning of the learned dimensions and reliance on GPT-based annotations that introduce cost, bias, and reproducibility concerns. Additional concerns include the heavy dependence on manually designed metrics, insufficient justification of PCA over alternative decorrelation or diversity mechanisms, and the fact that improvements, while consistent, are relatively modest given the added pipeline complexity.

Overall, although the paper shows promising directions and the authors engaged constructively with reviewers, the current version does not yet meet the bar for acceptance, and I therefore recommend rejection.

**Reviewer Concerns:**

Reviewer fDob: The rebuttal clarified the experimental setup, corrected figures and tables, added uncertainty estimates, and expanded evaluations across models and data budgets, largely resolving presentation and statistical concerns. It remains unclear **whether the gains are driven by PCA-based orthogonalization itself rather than broader candidate pools or multi-metric sampling.**

Reviewer nhUc: Concerns about scalability, thresholding, and generality were addressed through additional experiments, cost analysis, and clarification of the dataset and methodology. **The heavy reliance on GPT-based scoring** remains a central concern, as potential annotation bias is not cleanly separated from the reported improvements.

Reviewer KEDT:
All substantive concerns were resolved in the revision, leading the reviewer to raise their score.

Reviewer wRju: Missing baselines, scaling experiments, and cost–performance analyses were added, substantially strengthening the empirical evaluation. The paper still **does not conclusively show that PCA-based orthogonalization is uniquely responsible for the improvements.**

**Reviewer Scores:**

Reviewer KEDT: 4 --> 6
Reviewer wRju： 4 --> 6

Other socres would not be changed.

---

### Decision · Program_Chairs · 2026-01-26

Reject